# InfoMosaic-Bench: Evaluating Multi-Source Information Seeking in Tool-Augmented Agents

**Yaxin Du**[1]   **Yuanshuo Zhang**[1]   **Xiyuan Yang**[1]   **Yifan Zhou**[2]   **Cheng Wang**[1]
**Gongyi Zou**[4]   **Xianghe Pang**[1]   **Wenhao Wang**[3]   **Menglan Chen**[1]   **Shuo Tang**[1]
**Siheng Chen**[1]*
[1]Shanghai Jiao Tong University   [2]The Chinese University of Hong Kong   [3]Zhejiang University
[4]University of Oxford

## ABSTRACT

Information seeking is a fundamental requirement for humans. However, existing LLM agents rely heavily on open-web search, which exposes two fundamental weaknesses: online content is noisy and unreliable, and many real-world tasks require precise, domain-specific knowledge unavailable from the web. The emergence of the Model Context Protocol (MCP) now allows agents to interface with thousands of specialized tools, seemingly resolving this limitation. Yet it remains unclear whether agents can effectively leverage such tools—and more importantly, whether they can integrate them with general-purpose search to solve complex tasks. Therefore, we introduce InfoMosaic-Bench, the first benchmark dedicated to multi-source information seeking in tool-augmented agents. Covering six representative domains (medicine, finance, maps, video, web, and multi-domain integration), InfoMosaic-Bench requires agents to combine general-purpose search with domain-specific tools. Tasks are synthesized with InfoMosaic-Flow, a scalable pipeline that grounds task conditions in verified tool outputs, enforces cross-source dependencies, and filters out shortcut cases solvable by trivial lookup. This design guarantees both reliability and non-triviality. Experiments with 14 state-of-the-art LLM agents reveal three findings: (i) web information alone is insufficient, with GPT-5 achieving only 38.2% accuracy and 67.5% pass rate; (ii) domain tools provide selective but inconsistent benefits, improving some domains while degrading others; and (iii) 22.4% of failures arise from incorrect tool usage or selection, highlighting that current LLMs still struggle with even basic tool handling. Code is available here: `https://github.com/DorothyDUUU/Info-Mosaic`.

## 1 INTRODUCTION

Access to high-quality information is the fundamental driver of enhanced cognition, optimized decision-making, innovation, and societal progress. Each major advance in intelligent systems has been closely tied to progress in how they acquire and organize information: the advent of PageRank search engines (Page et al., 1999) made the web navigable at scale; the breakthrough of large language models (LLMs) (Brown et al., 2020; OpenAI, 2023; Kaplan et al., 2020) shifted information access from explicit retrieval to leveraging vast pre-trained knowledge; and most recently, web-search-augmented LLM agents (Nakano et al., 2021), such as various deep research product (OpenAI, 2025; Perplexity AI, 2025; Google, 2025), have transformed information seeking into an iterative process of querying, browsing, and synthesizing evidence. Already in wide use, these agents are now becoming indispensable, powering high-frequency workflows in science (Chai et al., 2025), business, and everyday decision-making (Shen, 2024).

Despite their growing adoption, today's agents remain fundamentally limited by their heavy reliance on open-web search. Online content is noisy, inconsistently formatted, and often unreliable (Wenzek et al., 2020; Vosoughi et al., 2018), making it insufficient for high-stakes applications. More impor-

---

*Corresponding Author: sihengc@sjtu.edu.cn

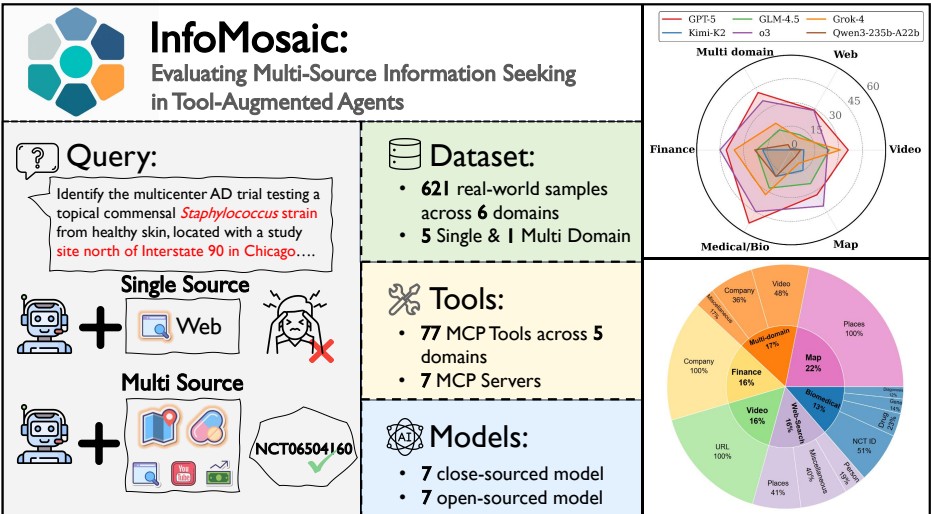

Figure 1: Overview of InfoMosaic-Bench. The benchmark evaluates multi-source information seeking in tool-augmented agents. (Left) Example query illustrating that single-source web search often fails, while multi-source tool use is required. (Center) Dataset statistics, including 621 samples across six domains, 77 MCP tools, and 14 models (7 closed- and 7 open-sourced). (Right) Radar plot showing domain-wise accuracy across models and the pie chart illustrating sample distribution across domains.

tantly, many real-world tasks require precise, verifiable, and domain-specific knowledge that web search simply cannot provide. For example, a financial analyst risks misleading conclusions without structured access to corporate filings and market data (Loughran & McDonald, 2011); a medical assistant cannot ensure patient safety without curated drug–side effect databases (U.S. Food and Drug Administration, 2022); and even route planning requires geospatial applications and transport schedules that cannot be recovered from fragmented web pages. These scenarios reveal a deeper challenge: general-purpose web search is not enough—reliable agents must integrate both general web information and specialized, domain-specific sources.

In parallel, the ecosystem of information-seeking tools is expanding rapidly. With the advent of the Model Context Protocol (MCP) tools (Hou et al., 2025), LLM agents can access thousands of heterogeneous data sources, ranging from biomedical databases (Flotho et al., 2025) to financial feeds (Zeng, 2025) and mapping services. Such advancements substantially enrich agent–environment interaction and broaden their information-seeking potential. While this shift appears to overcome the limitations of relying solely on general-purpose web search, it also raises two critical open questions: *(1) How effectively can LLM agents leverage domain-specific tools to access information within each individual field? (2) More importantly, can they seamlessly integrate general-purpose search with multiple specialized tools to tackle complex, multi-source information-seeking tasks?*

To answer these questions, we propose **InfoMosaic-Bench**, the first benchmark dedicated to evaluating the ability of LLM agents to perform multi-source information search using external tools. InfoMosaic-Bench comprises 621 synthesized tasks and 77 tools across six domains—medical/biology, finance, maps, video, web and multi-source seeking. This benchmark directly targets the two open challenges identified above and enables evaluation of both domain-specific tool usage and the harder setting of seeking of multi-source information. Unlike existing benchmarks, which either focus on generic web search in single source with single tool (like BrowseComp (Wei et al., 2025) and WebWalkerQA (Wu et al., 2025)) or correctness of isolated tool calls ($\tau$-Bench (Yao et al., 2024), MCP-Bench (Wang et al., 2025b)), InfoMosaic-Bench uniquely evaluates agents' ability to solve multi-source information-seeking tasks using contemporary and domain-specific MCP tools, with verified outputs ensuring reliability and non-triviality.

A key challenge in constructing such a benchmark is how to design tasks that inherently require multi-source search, rather than being solvable by a single tool or trivial web lookup. In practice, human curation has two limitations: no single author has broad cross-domain expertise, and crafting coherent multi-source tasks demands dozens of iterative tool calls, which is rarely sustainable by hand. Therefore, we propose **InfoMosaic-Flow**, an agentic data synthesis pipeline for multi-source information seeking task. The key idea is to leverage an organizer–workers architecture, where a single organizer acts as the commander, coordinating multiple domain-specific workers to enable scalable, cross-tool data synthesis. The organizer handles high-level planning, while each worker, tied to a particular domain, executes assigned tasks with its tools and returns precise results. This design enables integrative use of domain tools while maintaining robust reasoning, producing coherent and cross-tool grounded outputs. At last, we enforce multi-stage quality control combining automatic and carefully guided manual checks to guarantee reliability and difficulty.

After conducting extensive experiments, the results reveal three key findings: (1) *Web information alone is insufficient for precise domain reasoning:* even GPT-5 attains only 38.2% accuracy, showing that open-web search cannot meet the information needs of domain-specific tasks. (2) *Domain tools offer selective but limited benefits:* they improve performance in Map and Video but degrade in Medical, Finance, and Multi-domain, indicating that current agents are still far from being able to effectively exploit domain-specific tools within each field. (3) *Many failures come from incorrect domain-tool usage and selection:* nearly 22.4% of failures come from wrong tool usage and tool selection, demonstrating that agents lack the competence to reliably in even basic tool handling.

Looking forward, the benchmark exposes a fundamental gap: today's models excel at web search yet remain unable to reliably exploit domain tools or combine them effectively. Closing this gap is not a minor improvement, but a prerequisite for deploying trustworthy agents in high-stakes domains such as medicine, finance, and scientific discovery.

Our contributions are as follows:

- We identify the challenge that reliance on general-purpose web search is inadequate, and that no benchmark evaluates whether agents can leverage diverse domain-specific tools for reliable information seeking. We propose **InfoMosaic-Bench** to fill this gap.

- We propose **InfoMosaic-Flow**, an automated two-stage synthesis pipeline that grounds tasks in domain-wise tool evidence and refines them with web-based verification.

- Experiments show that relying on web search alone is insufficient for precise reasoning, while domain-specific tools can unlock additional capabilities, but current agents fail to robustly use them, leading to selective and inconsistent gains.

## 2 RELATED WORK

### 2.1 TOOL-USING LLMS

Early work on tool-augmented reasoning explored how to disentangle internal reasoning from external actions. ReAct (Yao et al., 2023a) pioneered this idea by interleaving chain-of-thought with explicit tool calls (e.g., search, calculator), enabling models to iteratively refine answers with external evidence. Building on this, Toolformer (Schick et al., 2023) showed that LLMs can self-supervise when and how to call APIs, while systems such as ToolLLM (Qin et al., 2023) and EasyTool (Yuan et al., 2024) and MCP-FLow (Wang et al., 2025a) scaled the breadth of API coverage and improved robustness of invocation. As LLM capabilities advanced, the focus shifted from invoking single APIs to handling long-horizon search and orchestration. Works such as Search-o1 (Li et al., 2025b), WebThinker (Li et al., 2025c), and R1-Searcher (Song et al., 2025) focus on persistent retrieval and orchestration in web search, highlighting the strengths and limitations of single-channel search-augmented reasoning. In contrast, the introduction of the Model Context Protocol (MCP) (Hou et al., 2025) expands tool use from web-only retrieval to a broad ecosystem of heterogeneous domain-specific tools, raising the new challenge of coordinating and integrating evidence across multiple sources. Our work targets precisely this gap, proposing a benchmark dedicated to evaluating multi-source information seeking in tool-augmented agents.

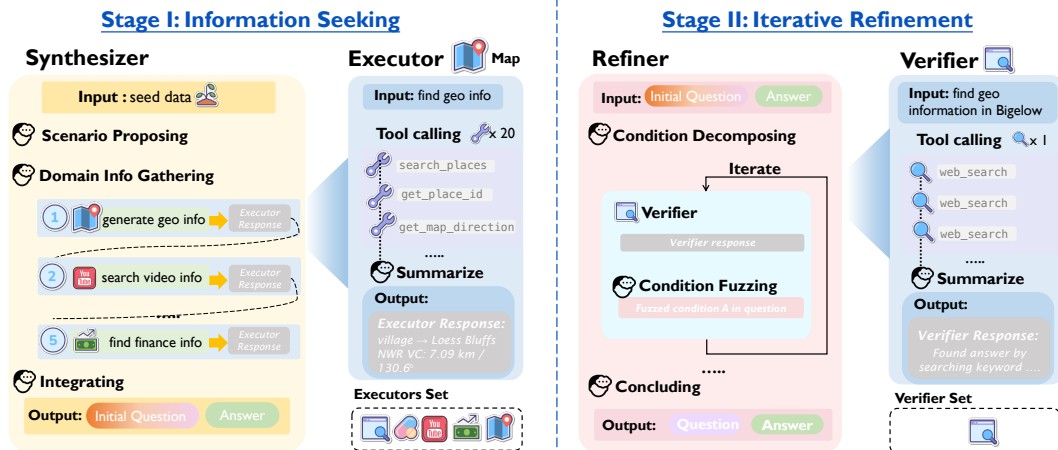

Figure 2: Overview of **InfoMosaic-Flow**. The synthesis pipeline is laid on an organizer–workers architecture, where a single organizer acts as the commander, coordinating multiple domain-specific workers. **Stage 1: Information Seeking** composing interdependent constraints and grounding them with verified multi-tool outputs to form initial QA pairs; **Stage 2: Iterative Refinement** revising drafts, pruning shortcuts, and enforcing multi-source reasoning.

## 2.2 BENCHMARKS FOR TOOL-USING AGENTS

There are three parallel lines of work for benchmark tool-augmented LLMs: 1) *API-centric benchmarks.* ToolBench(Qin et al., 2023) and related datasets (Patil et al.; Yao et al., 2024; Chen et al., 2025) evaluate an agent's ability to discover, select, and call APIs correctly. These efforts provide valuable coverage of API functionality and invocation robustness, but the evaluation typically centers on single-tool correctness rather than multi-source synthesis. 2) *Web/search-oriented benchmarks.* Datasets such as BrowseComp (Wei et al., 2025), WebWalkerQA (Wu et al., 2025), and MM-BrowseComp (Li et al., 2025a) evaluate agent's ability to engage with the open web, combining capabilities such as query reformulation, long-horizon reasoning, and information extraction from complex webpages. These benchmarks highlight the reasoning dimension of search-augmented agents and have advanced our understanding of how models operate in noisy and partially observable environments. However, the scope of tool use remains narrow: agents are restricted to web search and browsing, without evaluating whether they can coordinate multiple types of tools or integrate evidence beyond a single retrieval channel. 3) *MCP-style tool suites.* More recently, benchmarks have emerged around the MCP ecosystem, including MCP-Universe (Luo et al., 2025), MCP-Radar (Gao et al., 2025), MCP-Zero (Fei et al., 2025), and MCP-Bench (Wang et al., 2025b). These benchmarks expose agents to large-scale, heterogeneous tool environments and focus on aspects such as tool invocation correctness, execution robustness under complex tool spaces, or zero-shot tool discovery. However, they generally stop short of evaluating information seeking and long-horizon reasoning across tools. In short, none of them systematically evaluate whether LLM agents can reliably seek, combine, and reason over heterogeneous evidence sources. InfoMosaic-Bench fills this gap, providing a benchmark where every task is grounded in tool evidence and demands genuine multi-source reasoning.

## 3 METHODOLOGY

To construct a benchmark that reliably evaluates the ability of LLM agents to integrate evidence across multiple heterogeneous tools, we propose **InfoMosaic-Flow**, a scalable synthesis pipeline that generates tasks requiring non-trivial multi-source reasoning. This section introduces the overall design in Sec. 3.1 and describes our quality control procedure in Sec. 3.2.

## 3.1 Dataset Construction

Fig. 2 shows the overall generation with organizer-worker system. Specifically, an organizer is responsible for reasoning and formulating constraints/verification, while a worker is activated as a tool-calling event and follows instructions from the organizer to perform continuous tool calls and return consolidated evidence. This dual-agent designation has two advantages: (1) it isolates execution, preserving reasoning depth. The organizer handles decomposition or constraint reasoning; the worker handles fine-grained tool calls and evidence consolidation. This functional separation mitigates execution-induced noise in multi-step inference and reduces retrofitting of constraints to available tools. (2) It also expands exploration, improves multi-source coupling. The organizer remains tool-agnostic and selects only the target domain, while the executor freely chooses among all tools within that domain to satisfy the plan. This decoupling turns each subtask into a combinatorial search over the domain toolset, increasing tool diversity and coverage (see App. A.7).

Overall, the pipeline proceeds in two stages: (i) Information Seeking, the synthesizer composes interdependent constraints and the executor grounds them with verified outputs from multiple tools to form the initial QA pair; and (ii) Iterative Refinement, where drafts are repeatedly challenged and revised to prune single-source shortcuts, leaving only tasks that genuinely require multi-source reasoning. Fig. 2 illustrates the multi-domain instantiation: the other domains generation process simply by eliminating executor toolsets with medical/biology, finance, maps, video, and web, respectively.

### 3.1.1 Stage 1: Information Seeking

This stage takes seed data as input and generates a coherent multi-condition problem grounded in domain-wise evidence. The core idea of this stage is to use seed data to propose various scenarios that could diversify domain-wise tool callings to include different domain-specific information for problem synthesis. In addition, instead of relying on static templates or noisy web content, we actively query specialized tools and compose their retrieved information into tasks. This design guarantees that every problem is grounded in verifiable evidence and requires reasoning across multiple sources.

In this stage, the organizer in the agentic system is the synthesizer, and the worker is the executor with respective domain tools. The process of synthesizer works in the following steps: (1) **Scenario Proposing.** Starting from various seeds (e.g., Wikipedia or Baidu Baike, Qunar web, NCI IDs), the synthesizer proposes candidate scenarios that guide the construction of problems. This helps to provide diverse contexts that naturally invoke heterogeneous tools, increasing diversity beyond narrow or contrived tool-calling flows. (2) **Domain Information Gathering.** The synthesizer reasons step by step and emits high-level instructions (subtasks) that trigger tool calls, i.e. 'executor(subtask, domain)'. The executor, equipped with the domain toolset, selects and composes tools to retrieve verifiable domain-specific facts and returns organized evidence. The synthesizer consumes this evidence, updates the plan, and issues the next instruction. (3) **Integrating.** Finally, the synthesizer organizes the validated tool results into a coherent multi-sourced problem which requires multiple tool calls and cross-condition reasoning.

Overall, this design has two benefits. (i) First, by hiding tool internals, the synthesizer focuses on maintaining coherence and naturalness in the problem statement, rather than overfitting to tool quirks. (ii) Second, the information gathering loop enlarges the exploration space and includes diverse tools. As a result, Stage 1 ensures the generated problems are both coherent and inherently multi-source.

### 3.1.2 Stage 2: Iterative Refinement

While Stage 1 ensures that every synthesized problem is executable, the resulting tasks may still be trivial in practice. In particular, some problems can be answered by satisfying only a single clue, or even by issuing a generic web query without invoking multiple tools. Such cases do not reflect the real challenges faced by agentic systems, where reliable reasoning requires integrating evidence from multiple heterogeneous sources.

To eliminate these trivial cases, we introduce an iterative refinement stage. In this stage, the worker is a Verifier with only web-search tool, attempting to answer the Refiner-assigned problem through

web search. The refinement proceeds in three steps: (i) **Condition Decomposing**: the Refiner breaks down the synthesized problem into individual conditions and asks the Verifier to solve them independently; (ii) **Condition Fuzzing**: if any condition proves too revealing (e.g., directly exposing the answer via a single search), the Refiner rewrites, augments, or combines it with others to reduce shortcut solutions; (iii) **Concluding**: once no condition can independently yield the answer and the Verifier fails to solve the task via search alone, the refined set of conditions is recomposed into the final question.

The refinement process is repeated until two criteria are simultaneously met: (i) the problem cannot be solved by web search alone, and (ii) no single condition is sufficient to determine the answer. This ensures that each admitted question is both challenging and robust, demanding genuine multi-source reasoning. This refinement stage guarantees difficulty, since trivial shortcuts are removed and the remaining tasks genuinely require reasoning over multiple sources, which ensures that InfoMosaic-Bench provides a robust testbed for evaluating multi-source information seeking.

## 3.2 QUALITY CONTROL

To ensure the reliability of InfoMosaic-Bench, we adopt a series of quality control processes. We first apply two automated checks, including Tool-Call Filtering, Answer–Evidence Consistency, and Coherence Filtering, to remove trivial, noisy, or ill-formed tasks. After automatic filtering, we further conduct manual screening and revision by human annotators, who correct or discard problematic cases to improve factual alignment, coherence, and difficulty.

The automatic quality checks are as follows: (1)**Tool-Call Filtering.** We first enforce a minimum tool-call threshold in Stage 1, discarding samples below to eliminate under-constrained, low-seeking tasks and keep the benchmark focused on non-trivial multi-source reasoning. (2) **Answer–Evidence Consistency.** To guarantee traceability, we retain only items whose final answers are exactly derivable from collected tool outputs, ensuring every sample is grounded in verifiable information sources. (3) **Coherence Filtering.** We further remove tasks that exhibit incoherent or ill-formed conditions, such as contradictory constraints or unnatural phrasing, to guarantee that each problem maintains semantic coherence across its question, intermediate conditions, and final answer.

**Human Selection and Refinement.** After automatic filtering, we further review a sample for consistency, coherence, and difficulty (App. A.11.1); problematic items are revised or discarded. A dedicated user study (Sec. 5.4) further confirms reliability for evaluating multi-source seeking.

## 4 DATASET

**Statistics.** Table 1 summarizes the composition of InfoMosaic-Bench. The dataset contains 621 problems across 5 domains (medicine/biology, finance, maps, video, web), plus an additional set of explicitly cross-domain tasks. Each problem is paired with condition-level gold labels and traces of tool calls, enabling both final-answer evaluation and fine-grained diagnostic analysis. The benchmark is multilingual, containing 621 instances in total. Among these, 229 queries are written in Chinese (36.9%) and 392 in English (63.1%). Crucially, the evaluation pipeline (agent framework, tools, and metrics) is identical for both languages, as described in this section. In total, InfoMosaic-Bench incorporates 77 distinct tools spanning 7 servers, combined with condition-level supervision, ensuring that the benchmark provides a challenging and reliable testbed for evaluating multi-source information seeking. Detailed tool information are described in App. A.9.2.

Table 1: Domain distribution and key statistics of **InfoMosaic-Bench**. Refer to App. A.10 for more detailed statistics.

| Domain | Sample Number | MCP Tool Number |
|---|---|---|
| Medical/Biology | 83 | 15 |
| Web | 100 | 2 |
| Video | 100 | 11 |
| Finance | 100 | 29 |
| Map | 135 | 20 |
| Multi-Domain | 103 | - |
| **Total** | **621** | **77** |

**Evaluation Metrics.** We report both Accuracy and Pass Rate. **Accuracy** measuring strict end-to-end task success, reflecting whether the agent can complete information seeking and reasoning holistically. **Pass Rate**, in contrast, provides a more fine-grained view of agent performance based on

Table 2: Comparison of 14 LLM agents equipped with a web search tool on **InfoMosaic-Bench**, evaluated across six domains and the overall average. Metrics include Accuracy (**Acc**) and Pass Rate. The best overall Accuracy and Pass Rate is highlighted in bold.

| | Map | Medical/ Biology | Video | Web | Finance | Multi-domain | Overall Acc | Passrate |
|---|---|---|---|---|---|---|---|---|
| Close-Sourced Model | | | | | | | | |
| **GPT-5** | 32.59 | **53.10** | **36.00** | **29.00** | 41.00 | **41.75** | **38.18** | **67.48** |
| **o3** | **40.74** | 44.79 | 23.00 | 28.71 | **45.00** | 35.78 | 36.35 | 64.96 |
| **Grok-4** | 9.63 | 39.02 | 33.00 | 10.00 | 43.88 | 19.42 | 25.42 | 39.44 |
| **Claude-4.0-Sonnet** | 17.04 | 20.48 | 18.00 | 3.00 | 27.00 | 10.68 | 15.94 | 36.47 |
| **Qwen2.5-Max** | 7.41 | 7.23 | 5.00 | 0.00 | 9.00 | 1.94 | 5.15 | 15.72 |
| **Gemini-2.5-Flash** | 11.11 | 10.84 | 3.00 | 2.00 | 9.00 | 6.82 | 7.25 | 28.63 |
| **o4-mini** | 24.44 | 25.30 | 24.00 | 8.00 | 39.00 | 24.27 | 24.15 | 61.67 |
| Open-Weight Model | | | | | | | | |
| **GLM-4.5** | **24.44** | **27.71** | **24.00** | **11.00** | 22.00 | **14.56** | **20.61** | 26.98 |
| **Kimi-K2** | 14.81 | 19.28 | 8.00 | 1.00 | 18.00 | 0.00 | 10.14 | **39.72** |
| **Qwen3-235B-A22B** | 5.19 | 19.28 | 6.00 | 0.00 | 23.00 | 3.88 | 9.02 | 31.40 |
| **Qwen3-32B** | 8.15 | 8.43 | 4.00 | 1.00 | 23.00 | 1.94 | 7.73 | 33.80 |
| **DeepSeek-V3** | 9.63 | 7.23 | 1.00 | 0.00 | 16.00 | 2.91 | 6.28 | 25.40 |
| **Qwen3-Coder** | 9.63 | 4.82 | 6.00 | 0.00 | 9.00 | 1.71 | 5.44 | 19.25 |
| **Llama-4-Scout** | 0.74 | 4.82 | 0.00 | 0.00 | **22.00** | 2.91 | 4.83 | 21.03 |

associated test cases (subquestions with gold answers or subgoal checks, like many coding tasks Du et al. (2025)).

## 5 EXPERIMENTS

### 5.1 EXPERIMENTAL SETUP

**Models Evaluated.** In our experiments, we evaluate 7 closed-source and 7 open-source LLMs. Details of these LLMs can be found in Table 5.

**Agent Framework.** For the agent framework, we adapt the most popular framework ReAct (Yao et al., 2023b) equipped with OpenAI's tool-calling interface (OpenAI, 2025) and a Python Sandbox (Pang et al., 2025) from which LLMs receive the tool execution results. More details can be found in Sec. A.4.2.

**Evaluation.** As introduced in Sec. 4, we use Accuracy (Acc) and Pass Rate (PR) as our evaluation metrics. Instead of relying solely on exact match, we leverage an LLM to judge whether the predicted answers align with the references, which alleviates cases where semantically correct outputs cannot be captured by string matching. The detailed evaluation prompts are provided in the App. A.12.7.

### 5.2 MAIN RESULTS

**InfoMosaic-Bench demonstrates that web search alone is insufficient for multi-source reasoning.** Table 2 reports results for 14 state-of-the-art LLM agents limited to a web-search tool. We observe that: (1) Current agent system performs poorly on this task. Even the best closed-source model (GPT-5) attains only 38.2% accuracy and a 67.5% pass rate. (2) Closed-source models exceed open-source ones by 15–20% on accuracy, yet both are constrained by web information. (3) Pass rates consistently surpass exact accuracy, reflecting that agents often satisfy some conditions but fail to integrate all of them into a correct final answer.

**InfoMosaic-Bench reveals stark differences across domains, especially in video, map, and multi-domain.** The bottom-right radar in Fig.1 summarizes domain-wise accuracy for six models on InfoMosaic-Bench. We find that (1) agents with only web search are highly uneven across domains: best scores reach 53% (Medical/Biology) and 45% (Finance) but drop to 36% (Video)

Table 3: Comparison between web-only and domain-tool agent accuracy across six domains and the overall average for GLM-4.5 and GPT-5. The colored numbers indicate the difference between domain-tool and web-only settings. Values in green denote accuracy improvements when domain tools are used, while values in red denote decreases.

| | Tool | Map | Medical/Bio | Video | Web | Finance | Multi-domain | Overall |
|---|---|---|---|---|---|---|---|---|
| **GLM-4.5** | web | 24.44 | 27.71 | 24.00 | 11.00 | 22.00 | 14.56 | 20.61 |
| | domain | 30.37+5.93 | 34.94+7.23 | 25.00+1.00 | 7.00-4.00 | 20.00-2.00 | 12.62-1.94 | **21.51**+0.90 |
| **GPT-5** | web | 32.59 | 53.10 | 36.00 | 29.00 | 41.00 | 41.75 | 38.18 |
| | domain | 40.00+7.41 | 43.37-9.73 | 46.00+10.00 | 32.00+3.00 | 30.00-9.00 | 39.81-1.94 | **38.61**+0.43 |

and 40.74% (Map). (2) Capabilities vary by domain—for example, Grok-4 does relatively well on Video, whereas GPT-5 struggles there.

## 5.3 ANALYSIS

### 5.3.1 DOMAIN TOOL ANALYSIS

**Comparison with only web search tool.** To evaluate how LLMs perform with domain-specific tools, we conduct experiments under two settings. For each single-domain evaluation, the agent is provided only with that domain's specialized tools (for multi-domain, all tools are provided). We report answer accuracy for GLM-4.5 and GPT-5 under only web search tool and domain-tool settings in Table 3. We find that: (1) On average, domain tools yield only marginal gains, indicating that the bottleneck is not tool availability but tool use—how agents plan, select, parameterize, and time their calls. (2) Both GPT-5 and GLM-4.5 see clear gains in map and video domains because these tasks depend on structured, exclusive signals (e.g., spatial queries, video metadata) that web search cannot reliably provide. (3) Accuracy drops on multi-domain tasks with many tools, highlighting cross-source orchestration issues: selecting and chaining tools raises planning complexity and error propagation.

**Tool use result analysis.** To analyze the reason why the accuracy will drop in certain domains. We collect and categorize the results of tool calls into four types: usage error (wrong function calling), selection error (wrong tool selections), invalid result (successful but irrelevant/unhelpful calling), and valid result (successful and useful tool calling). Fig.4a shows the distributions of these types in each domain. We have the following findings: (1) As shown by the line, better tool usage yields more useful information and leads to stronger model performance. (2) Tool usage error rate correlates with tool complexity. Bio and Multi-domain with larger parameter numbers exhibit higher usage-error rates. Finance and Multi-domain host the largest toolsets and show markedly higher selection error rates, implying larger tool inventories increase selection risk. (3) Most tool results are unhelpful and contribute little to answering the question.

### 5.3.2 SCALING ANALYSIS

We analyze the relationship between the number of tool calls and performance. Fig. 3a and 3b report results of GLM-4.5 across three representative domains (Finance, Bio, and Video), showing how accuracy and pass rate change with the number of tool calls. Fig. 3c compares 4 different models, indicating the growth of input token length as the number of tool calls increases. We find that: (1) Acc and PR generally increase with more tool calls. (2) Performance plateaus after 8 tool calls, with further tool calls sometimes reducing accuracy and pass rates due to redundant rather than useful information. (3) In Fig.3c, input token length rises quickly with the number of tool calls until a turning point, after which extra calls add little context. This turning point reflects each model's effective tool-usage limit. Across models, it correlates with overall accuracy ($R^2 = 0.57$), indicating a moderate positive link between effective tool-call capacity and task performance.

### 5.3.3 FAILURE MODE ANALYSIS

Fig. 4b shows GPT-5's web-only failure distribution on InfoMosaic-Bench using a six-class, primary-cause label (Retrieval Miss, Tool Misuse, Reasoning Gap, Confirmation Bias, Overgeneralization, Context Misread) which is detailed in Appendix A.5. Retrieval Miss (39.6%) and Over-

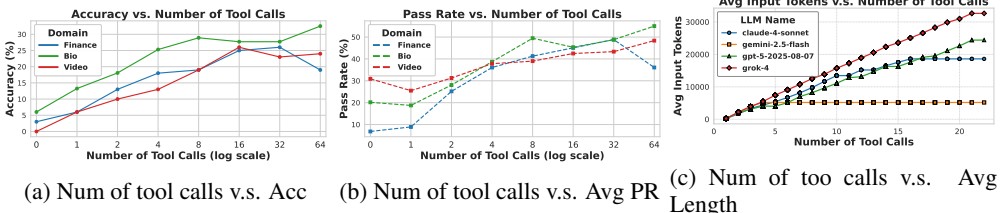

(a) Num of tool calls v.s. Acc    (b) Num of tool calls v.s. Avg PR    (c) Num of too calls v.s. Avg Length

Figure 3: Relationship between performance and input length. (a) and (b) show results of GLM-4.5's Acc and PR across Finance, Bio, and Video domains. (c) compares different models, showing average input token length against the number of tool calls.

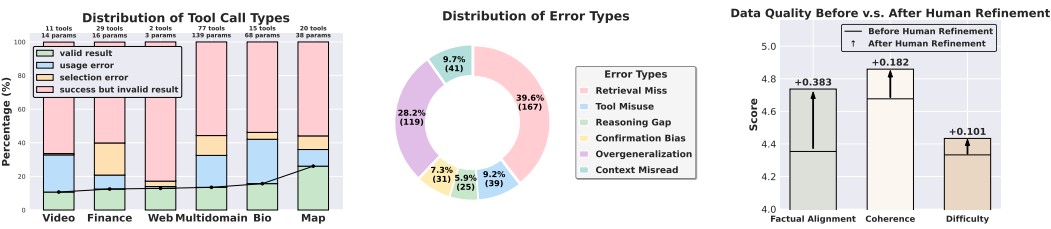

(a) Distribution of tool-call result types in 6 domains.    (b) Distribution of 6 error types only using the web search.    (c) Comparison of human evaluation before and after refinement.

Figure 4: Distribution analysis. Fig. 4a shows the distributions of 4 tool calling result categories in 6 domains, Fig. 4b shows 6 failure modes of GPT-5 with only web search, and Fig. 4c shows improvements after refinement.

generalization (28.2%) dominate, indicating failures stem mainly from retrieval and evidence selection rather than final-step reasoning—underscoring the need for domain tools and stronger search orchestration.

## 5.4 HUMAN STUDY

We conduct a human study on 120 randomly sampled problems across domains: three NLP-trained graduate annotators independently rated pre- and post-check versions on factual alignment, coherence, and task difficulty (guidelines in Appendix A.11.1). Fig.4c shows (1) high pre-check scores, indicating strong baseline quality from tool-grounded synthesis; and (2) the largest post-check gain in factual alignment, correcting evidence–answer mismatches. Agreement is high (Cohen's $\kappa = 0.92$), confirming reliability.

## 6 CONCLUSION

In this work, we propose *InfoMosaic-Bench*, the first benchmark in evaluating multi-source information seeking in tool-augmented agents, spanning 6 domains with 77 heterogeneous tools. To construct domain-wise and cross-domain information seeking tasks, we also proposed *InfoMosaic-Flow*, a scalable synthesis methodology. Our extensive experiments demonstrate that (i) web search alone is insufficient for domain-specific information seeking tasks, (ii) current agents remain disproportionately better at web search than at leveraging domain-specific tools. We expect InfoMosaic-Bench to catalyze a shift from web-only search to principled, auditable multi-tool information seeking, accelerating progress on high-stakes domains such as finance and science. Future work may extend the synthesis pipeline to additional modalities, interactive environments, pushing LLM agents closer to real-world deployment.

## 7 ETHICS STATEMENT

Our paper fully complies with the ICLR Ethics guidelines. Specifically, the research is designed to contribute positively to society and human well-being by proposing InfoMosaic-Bench, the first benchmark dedicated to multi-source information seeking in tool-augmented agent, with careful consideration of potential risks and mitigation strategies to avoid harm. We make sure our experiment results are reproducible and our code and dataset are accessible. This study does not involve any trade secrets, client data, non-public business strategies, financial information, research data, pre-publication scholarly articles, and patent applications. The data used for the synthesis are all publicly available, and citations have been added in the appendix. We comply with the corresponding licenses to ensure respect for the original sources. All human labor involved in this paper has been conducted with full respect for the workers and creators. Since our data come from real-world sources, there is no issue of bias from human annotators. Our research does not involve any harm or discriminatory practices.

## 8 REPRODUCIBILITY STATEMENT

Our code and dataset are available here `https://github.com/DorothyDUUU/Info-Mosaic`. Our approach and experiments were designed with reproducibility in mind, and we encourage others to build upon our work using the provided resources.

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

# A APPENDIX

## A.1 THE USE OF LARGE LANGUAGE MODELS (LLMS)

We acknowledge the use of large language models (LLMs) in the this work. Specifically, GPT-5 was employed for language polishing and refinement of the manuscript, while GPT-4o was used as a tool for evaluation and error analysis. The LLMs did not contribute to research ideation, methodology development, or experimental execution.

## A.2 USED DATA SOURCE

Table 4 lists the data sources used to generate seed data and seeking information. We adhere to all knowledge-use policies of these websites and products.

Table 4: List of data sources.

| Name | Website | Type |
|---|---|---|
| Wikipedia | www.wikipedia.org | Seed data source |
| Baidu Baike | baike.baidu.com | Seed data source |
| Qunar | www.qunar.com | Seed data source |
| NORD | rarediseases.org | Seed data source |
| ClinicalTrials.gov | clinicaltrials.gov | Seed data source |
| AMap MCP | github.com/sugarforever/amap-mcp-server | MCP server |
| Google Map MCP | github.com/cablate/mcp-google-map | MCP server |
| Bio MCP | github.com/genomoncology/biomcp | MCP server |
| YouTube MCP | github.com/jikime/py-mcp-youtube-toolbox | MCP server |
| FMP MCP | github.com/cdtait/fmp-mcp-server | MCP server |

## A.3 PROBLEM FORMULATION

**Notation.** Let the domain be $\mathcal{D}$, the queries in $\mathcal{D}$ be $\mathcal{Q}_\mathcal{D}$, and the user query be $q \in \mathcal{Q}_\mathcal{D}$. Let the set of available tools be

$$\mathcal{T}_{\text{avail}} = \{\mathtt{T}_1, \ldots, \mathtt{T}_m\},$$

where each tool $\mathtt{T}_i$ is specified by interface metadata (name, description, parameter schema, output schema). Let $\mathtt{GT}$ denote the ground-truth answer to $q$ and $K$ be the max tool calling limit. A task instance denoted as

$$\tau = (q, \mathcal{T}_{\text{avail}}, K, \mathtt{GT}).$$

**Evaluation.** Let $\mathcal{M} = \{M_1, M_2, \ldots, M_n\}$ denote the set of LLMs and $\mathcal{A} = \{A_1, A_2, \ldots, A_p\}$ the set of agent frameworks. For a given $(M, A) \in \mathcal{M} \times \mathcal{A}$ and task $\tau$, the interaction generates a message history

$$H = (h_1, h_2, \ldots, h_T),$$

where each $h_t = (r_t^{(M,A)}, r_t^{\text{tool}})$ records the responses from agents and any tool invocations. The evaluation function

$$E : (H, \tau) \rightarrow \{0, 1\}$$

assigns 1 if the user query is correctly solved under predefined success criteria, and 0 otherwise. Correctness is determined by automated checks (e.g., verifying output content or exactly matching). Details of evaluation could be found in Sec. 5.1.

## A.4 EXPERIMENT SETUP

### A.4.1 MODELS

We evaluate a broad spectrum of current top tier LLMs, including both close-sourced and open-sourced models. Table 5 summarizes their key features such as model size, release date, openness, and access links. Through the evaluation of these latest state-of-the-art LLMs, we ensure the validity and reliability of our main conclusions.

Table 5: Details about evaluated LLMs.

| Model | Size | Release Date | Status | Link |
|---|---|---|---|---|
| GPT-5 | — | 2025-08-07 | Closed | https://platform.openai.com/docs/models/gpt-5 |
| Grok-4 | — | 2025-07-09 | Closed | https://x.ai/news/grok-4 |
| Claude-4.0-Sonnet | — | 2025-05-23 | Closed | https://www.anthropic.com/claude/sonnet |
| o3 | — | 2025-04-16 | Closed | https://platform.openai.com/docs/models/o3 |
| o4-mini | — | 2025-04-16 | Closed | https://platform.openai.com/docs/models/o4-mini |
| Qwen2.5-Max | — | 2025-01-25 | Closed | https://qwenlm.github.io/zh/blog/qwen2.5-max/ |
| Gemini-2.5-flash | — | 2025-06-17 | Closed | https://deepmind.google/models/gemini/flash/ |
| GLM-4.5 | 355B | 2025-07-28 | Open | https://huggingface.co/zai-org/GLM-4.5 |
| Qwen3-235B-A22B | 235B | 2025-04-29 | Open | https://huggingface.co/Qwen/Qwen3-235B-A22B |
| Qwen3-32b | 32.8B | 2025-04-29 | Open | https://huggingface.co/Qwen/Qwen3-32B |
| Llama-4-Scout | 109B | 2025-04-05 | Open | https://huggingface.co/meta-llama/Llama-4-Scout-17B-16E |
| DeepSeek-V3 | 685B | 2025-03-24 | Open | https://huggingface.co/deepseek-ai/DeepSeek-V3-0324 |
| Kimi-K2 | 1T | 2025-07-11 | Open | https://huggingface.co/moonshotai/Kimi-K2-Instruct |
| Qwen3-Coder | 30.5B | 2025-07-22 | Open | https://huggingface.co/Qwen/Qwen3-Coder-30B-A3B-Instruct |

### A.4.2 AGENT FRAMEWORK

We build our agent framework based on ReAct (Yao et al., 2023a), which serves as the foundation of the agent's reasoning and actions. The framework integrates a Python Sandbox and follows the OpenAI function calling interface (OpenAI, 2025), enabling the LLM to invoke external tools and consume their outputs in a structured manner.

Concretely, we implement multi-turn interactions where tool metadata is serialized into JSON Schema format and provided to the LLM through the function calling interface. The LLM responds with a structured tool call request, which we automatically translate into a Python code snippet. This code is then executed inside the Python Sandbox (Pang et al., 2025), and the execution results (standard outputs or errors) are captured. Finally, the returned results are appended to the dialogue history and passed back to the LLM as additional context for subsequent reasoning steps. We set the tool calling limit to 20, ensuring that the agent terminates tool calls after repeated unsuccessful attempts to solve the problem.

This design allows the framework to (i) support multiple domain-specific tools in a uniform schema, (ii) enforce execution safety through sandbox isolation, and (iii) tightly couple the reasoning trace of the LLM with actual tool outputs.

### A.5 DETAILS OF FAILURE MODES ANALYSIS

We define six categories of failure: (1) Retrieval Miss, where the agent fails to extract key information already present in the tool or knowledge base, often producing "cannot determine" answers or overlooking relevant facts; (2) Tool Misuse, where the agent misinterprets tool outputs or provides incorrect parameters or commands, leading to irrelevant, distorted, or erroneous results (e.g., miscalculations); (3) Reasoning Gap, where correct information fragments are retrieved but not coherently linked, resulting in logical jumps, causal confusion, or inconsistent conclusions; (4) Confirmation Bias, where the agent selectively emphasizes evidence supporting its initial hypothesis while downplaying or ignoring contradictory tool outputs; (5) Overgeneralization and Hallucination, where insufficient evidence leads to unfounded guesses or fabricated details not supported by tool results; and (6) Instruction and Context Misunderstanding, where the agent misinterprets the user's intent or task context, producing answers that may be partially correct but irrelevant to the actual query.

### A.6 T-SNE VISUALIZATION OF QUERY EMBEDDINGS

Fig. 5 reveals clear domain-level clustering of query embeddings, indicating strong semantic separability across domains. Limited overlap suggests occasional cross-domain ambiguity, while dispersion reflects intra-domain diversity.

### A.7 VALIDATING THE SYNTHESIS PIPELINE

We perform ablations to verify the necessity of the two core components of InfoMosaic-Flow: web-based verification and planner–executor interaction.

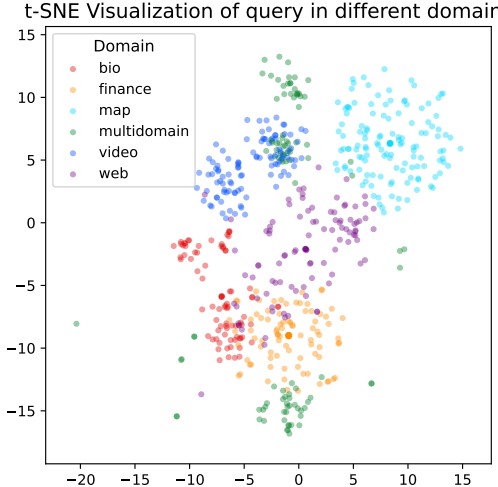

Figure 5: t-SNE visualization of query embeddings across domains. Each point is a query, colored by domain.

Table 6: Effect of planner–executor design. Removing the executor collapses tool usage and reduces task complexity.

| Executor Setting | Avg. Tool Calls | Avg. Tools Used |
|---|---|---|
| With Executor | 59.1 | 43.1 |
| Without Executor | 7.8 | 5.6 |

**Effect of Planner–Executor Interaction.** The default pipeline employs GPT-5 as the planner and executor pair. As shown in Table 6, removing the executor collapses the synthesis space. With executor, synthesized tasks involve on average 59.1 tool calls across 43.1 unique tools. Without executor, this drops sharply to 7.8 calls across 5.6 tools, indicating severe loss of heterogeneity and task richness.

**Effect of Web-Evolving Verification.** Stage-2 verification is implemented with GPT-5 acting as both the main synthesizer and executor. Table 7 show that without this step, many tasks collapse to trivial single-source queries, yielding inflated accuracy (45.1%). To test pruning strategies, we substitute the executor with GPT-5. In Quick Fuzz mode, the GPT-5 executor rewrites conditions without invoking web tools, modestly reducing shortcuts but leaving residual triviality (39.7%). Advanced Fuzz applies iterative probing and constraint rewriting, lowering accuracy to 31.3%, which indicates stronger resistance to shortcuts and closer alignment with true multi-source reasoning.

Web-evolving verification prevents trivial shortcuts, and planner–executor interaction ensures sufficient task richness. Both are indispensable for generating a benchmark that reflects realistic multi-source reasoning challenges.

## A.8 AGENT PROBLEM-SOLVING ANALYSIS

### A.8.1 PATTERNS ANALYSIS IN SEARCHING PROBLEMS

After analyzing the trajectories of Agents solving dataset problems, we can find that advanced Agents with higher evaluation scores can already exhibit clear, structured thought processes and steps, which are not prompted by humans. Based on our analysis, GPT-5 demonstrates a specific, sequential long-range "Searching-Reasoning-Evaluating" trajectory when solving problems in the map domain, which is: "**Broad Search → Targeted Information Retrieval → Solution Evaluation → Response Calibration**".

Table 7: Effect of web-evolving verification. Without pruning, many tasks collapse to trivial web lookups, inflating accuracy. Advanced fuzzing most effectively suppresses shortcuts.

| Verification Setting | Accuracy (%) |
|---|---|
| w/o Condition Pruning | 45.1 |
| Quick Fuzz (GPT-5, no web) | 41.7 |
| Advanced Fuzz (GPT-5, web) | 31.3 |

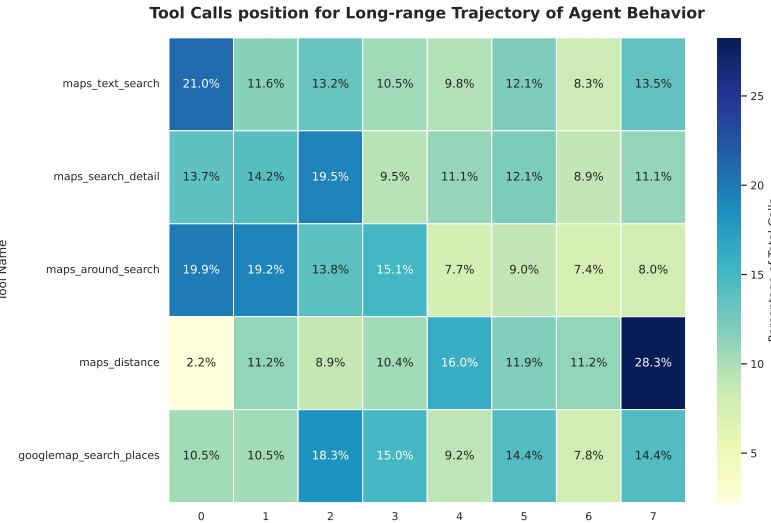

Figure 6: Heatmaps for sequence of tool calls made by the GPT-5 when solving map domain problems.

**Broad Search**: In the initial stage, the agent autonomously chooses and prefers to use generalized keyword searches, such as `maps_text_search` and `maps_around_search`, which retrieve keywords and return a larger number of candidate answers and their accurate place IDs. The main purpose of this step is to include the correct answer through a generalized search, thereby constraining and narrowing down the broad search space into a list of choices.

**Targeted Information Retrieval**: After searching for candidate answers, the agent proactively invokes more fine-grained search tools to perform a deep, targeted search for specific POI. For example, agent may call functions `maps_search_detail` with an accurate ID provided by broad-search tools or functions named `maps_distance` for targeted information seeking. The use of these tools enables the model to conduct precise searches on the candidate answers, allowing it to further filter out unreasonable options and constrain the search space to a smaller set of candidates.

**Solution Evaluation**: After filtering, the model is left with 3 to 4 highly similar candidate answers. At this point, in addition to using deep search tools, the model integrates all the information to perform its final reasoning and selection.

**Response Calibration**: Eventually, agent call tools to validate the candidate answer.

Fig. 6 shows the frequency of making diverse tool calls when solving map problems. We calculate the frequency of invoked tools in tool-call sequence of GPTT-5 and. We segment the action trajectory into 8 relative positions, representing different stages of the entire thought process. It is obvious that the agent prefers to call broad search tools (`maps_text_search` and `maps_around_search`) mostly at the beginning of the action trajectory. Then, deep and targeted search tools including `maps_search_detail` and `google_maps_search_places` begin to emerge. At last, in the validation, more tool calls of `maps_distance` are used for checking the candidate answer.

### A.8.2 ERROR STEPS ANALYSIS

Based on our segmentation of agent's tool-calling trajectory, we further investigated the specific step where the agent's error occurred. The results are shown in Fig. 7. We find that agents are most prone to errors at **Broad Search**, namely when the agent fails to retrieve the correct answer and add it to the candidate list, especially when agent is performing keyword searching. Improper keywords composition may lead to an incorrect or empty search result hit.

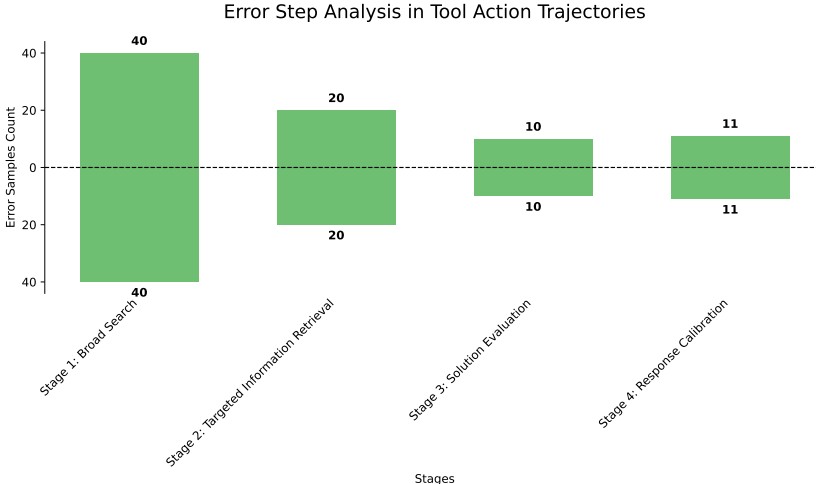

Figure 7: Distribution of agent errors across the segmented steps of the tool-calling trajectory

The following example[1] demonstrates how the agent's use of incorrect keyword combinations (either too complex or too generalized) resulted in the tool's search failing to return a hit. In this example, too restritive searching keywords lead to empty response.

```
{
    "function": {
      "arguments": "{"query": "Yuecheng District, Shaoxing, Scenic
          spots and stone bridges near Changqiao Zhijie"}",
      "name": "googlemap_search_places"
    },
    "type": "function"
}
{
    "role": "tool",
    "content": "{'tool_result': 'The Basic information for the query:
        Yuecheng District, Shaoxing, Scenic spots and stone bridges
        near Changqiao Zhijie with 0 results'}"
}
```

### A.9 REPRESENTATIVE BENCHMARK INSTANCES

### A.9.1 EXAMPLES

The following examples illustrate instances from various domains in our dataset, showcasing how queries are structured with associated testcases and ground truths (GT). In cases where a testcase condition can be decomposed into a subquestion:subanswer format, the corresponding GT is provided and is not null, ensuring verifiable details. However, if a condition lacks an explicit subanswer (e.g., due to its nature as a direct factual check without granular breakdown), it is split into multiple independent testcases for thorough inspection and validation.

---

[1]Model response in Chinese, translated in English

| Bio Domain | |
| --- | --- |
| **Query** | A clinical trial aimed to assess the impact of a RET inhibitor on cardiac parameters in healthy adults. Which NCT number identifies the trial with these features? 
 1. The trial utilized a thorough QT/QTc (TQT) design with a randomized, double-blind, four-period crossover approach, evaluating two doses of the study drug against placebo and moxifloxacin as a positive control. 
 2. Eligibility required serum potassium $\geq$ 3.8 mEq/L, calcium $\geq$ 8.5 mg/dL, and magnesium $\geq$ 2.0 mEq/L at screening, along with a BMI of $18$–$32\,\mathrm{kg/m^2}$. 
 3. The primary endpoint involved model-predicted, placebo-adjusted QTcF interval changes ($\Delta\Delta$QTcF) measured via 12-lead ECG over a 24-hour period post-dose. 
 4. Conducted at a single site in Tempe, Arizona, the trial's primary phase concluded in June 2019, with results first published in September 2025. |
| **Testcase (Condition)** | The trial utilized a thorough QT/QTc (TQT) design with a randomized, double-blind, four-period crossover approach, evaluating two doses of the study drug against placebo and moxifloxacin as a positive control. |
| **Testcase (Ground_truth)** | Single-dose, randomized, double-blind, placebo- and positive-controlled, 4-way crossover; treatments included selpercatinib 320 mg, selpercatinib 640 mg, placebo, and 400 mg moxifloxacin (positive control). |
| **Testcase (Condition)** | Eligibility required serum potassium $\geq$ 3.8 mEq/L, calcium $\geq$ 8.5 mg/dL, and magnesium $\geq$ 2.0 mEq/L at screening, along with a BMI of $18$–$32\,\mathrm{kg/m^2}$. |
| **Testcase (Ground_truth)** | Exclusion if $K^+$ < 3.8 mEq/L, $Ca^{2+}$ < 8.5 mg/dL, $Mg^{2+}$ < 2.0 mEq/L; BMI $\geq$ 18.0 and $\leq 32.0\,\mathrm{kg/m^2}$. |
| **Testcase (Condition)** | The primary endpoint involved model-predicted, placebo-adjusted QTcF interval changes ($\Delta\Delta$QTcF) measured via 12-lead ECG over a 24-hour period post-dose. |
| **Testcase (Ground_truth)** | Primary outcome: placebo-corrected change from baseline in QTcF ($\Delta\Delta$QTcF) based on model-predicted effect; 12-lead ECG extracted from continuous recordings at pre-dose and up to 24 hours post-dose. |
| **Testcase (Condition)** | Conducted at a single site in Tempe, Arizona, the trial's primary phase concluded in June 2019, with results first published in September 2025. |
| **Testcase (Ground_truth)** | Site: Celerion, Tempe, Arizona, United States; Primary completion date: 2019-06-21; Results first posted: 2025-09-18. |
| **GT (NCT number)** | **NCT05630274** |

Table 8: Example instance from the Bio domain.

| Map Domain | |
|---|---|
| **Query** | 在大连市中山区，寻找一个符合以下特征的地标： 1. 在行政区划上属于中山区（区划代码210202） 2. 从青泥洼桥地铁站步行至此约需31分59秒，步行距离约2.40公里 3. 从大连站驾车到此全程约4.22公里，预计用时约12分59秒；而从大连站乘坐地铁5号线至劳动公园站并步行到达的公共交通方案总用时约42分22秒，因此驾车更快超过5分钟 该地标的名称是？ / *[Translation: In Zhongshan District of Dalian, find a landmark that meets the following criteria: (1) It administratively belongs to Zhongshan District (division code 210202); (2) Walking from Qingniwaqiao Metro Station to this place takes about 31 minutes 59 seconds, with a walking distance of about 2.40 km; (3) Driving from Dalian Railway Station to this place covers about 4.22 km and is estimated to take about 12 minutes 59 seconds, whereas taking Metro Line 5 from Dalian Railway Station to Laodong Park Station and then walking takes about 42 minutes 22 seconds in total, so driving is more than 5 minutes faster. What is the name of this landmark?]* |
| **Testcase (Condition)** | 在行政区划上属于中山区（区划代码*210202*） / [**Translation**: Administratively belongs to Zhongshan District (division code 210202).*]* |
| **Testcase (Ground_truth)** | `null` |
| **Testcase (Condition)** | 从青泥洼桥地铁站步行至此约需*31*分*59*秒，步行距离约*2.40*公里 / *[**Translation**: Walking from Qingniwaqiao Metro Station to this place takes about 31 minutes 59 seconds, with a walking distance of about 2.40 km.]* |
| **Testcase (Ground_truth)** | `null` |
| **Testcase (Condition)** | 从大连站驾车到此全程约4.22公里，预计用时约12分59秒；而从大连站乘坐地铁5号线至劳动公园站并步行到达的公共交通方案总用时约42分22秒，因此驾车更快超过5分钟 / *[**Translation**: Driving from Dalian Railway Station to this place covers about 4.22 km and is estimated to take about 12 minutes 59 seconds; in contrast, travelling from Dalian Railway Station by Metro Line 5 to Laodong Park Station and then walking takes about 42 minutes 22 seconds in total, so driving is more than 5 minutes faster.]* |
| **Testcase (Ground_truth)** | `null` |
| **GT (Toponym)** | 大连观光塔 / *[Translation: Dalian Sightseeing Tower]* |

Table 9: Example instance from the Map domain.

**Finance Domain**

| | |
|---|---|
| **Query** | Identify a publicly traded technology company satisfying the following criteria: (1) Headquarters located in Austin, Texas; (2) Positive net income reported in Q1 2025 (quarter ending March 31, 2025), but negative net income in Q2 2025 (quarter ending June 30, 2025); (3) Year-to-date stock price change as of September 19, 2025, between 10% and 20%, with beta coefficient exceeding 5.0; (4) Q2 2025 revenue (quarter ending June 30, 2025) between $70 million and $80 million USD. |
| **Testcase (Condition)** | *Its headquarters is located in Austin, Texas.* |
| **Testcase (Ground_truth)** | `Austin, Texas` |
| **Testcase (Condition)** | *It reported positive net income in Q1 2025 (quarter ending March 31, 2025) but negative net income in Q2 2025 (quarter ending June 30, 2025).* |
| **Testcase (Ground_truth)** | `Q1 2025 net income: $580,693,000 (positive); Q2 2025 net income: -$936,799,000 (negative)` |
| **Testcase (Condition)** | *Its year-to-date stock price change as of September 19, 2025, was between 10% and 20%, and its beta coefficient exceeds 5.0.* |
| **Testcase (Ground_truth)** | `YTD change as of 2025-09-19: 15.68%; Beta: 6.6067` |
| **Testcase (Condition)** | *Its Q2 2025 revenue (quarter ending June 30, 2025) was between $70 million and $80 million USD.* |
| **Testcase (Ground_truth)** | `$78,628,000` |
| **GT (Stock Symbol)** | **CORZ** |

Table 10: Example instance from the Finance domain.

**Video Domain**

| | |
|---|---|
| **Query** | On YouTube, there is a video that meets all of the following conditions: 1) Uploaded in September 2025; 2) Duration is approximately 2 minutes and 36 seconds; 3) From a channel with over 1.7 million subscribers; 4) A top comment with over 1,000 likes mentions both "Earthquake" and "Baby scene". Which video is this? Provide its URL.. |
| **Testcase (Condition)** | *Uploaded in September 2025* |
| **Testcase (Ground_truth)** | `null` |
| **Testcase (Condition)** | *Duration is approximately 2 minutes and 36 seconds;* |
| **Testcase (Ground_truth)** | `null` |
| **Testcase (Condition)** | *From a channel with over 1.7 million subscriber* |
| **Testcase (Ground_truth)** | `null` |
| **Testcase (Condition)** | *A top comment with over 1,000 likes mentions both "Earthquake" and "Baby scene"* |
| **Testcase (Ground_truth)** | `null` |
| **GT (URL)** | **https://www.youtube.com/watch?v=Rev9xjajSlM** |

Table 11: Example instance from the Video domain.

| Web Domain | |
|---|---|
| **Query** | In a transoceanic venture undertaken by two vessels, the smaller craft had a commander with a rank-style title, while another man actually held the practical authority for navigation. The sailing schedule slipped because that man went inland to collect debts, and much later a regional museum hosted a public lecture centered on him. Who was he? |
| **Testcase (Condition)** | *In a transoceanic venture undertaken by two vessels* |
| **Testcase (Ground_truth)** | `Ark and Dove` |
| **Testcase (Condition)** | *the smaller craft had a commander with a rank-style title* |
| **Testcase (Ground_truth)** | `Captain Wintour` |
| **Testcase (Condition)** | *Historic St. Mary's City (HSMC)* |
| **Testcase (Ground_truth)** | `Richard Orchard` |
| **GT (Person Name)** | **Richard Orchard** |

Table 12: Example instance from the Web domain.

| Multi-Domain | |
|---|---|
| **Query** | Identify the stock symbol of a U.S.-listed company fulfilling all criteria: (1) Headquarters in a U.S. city renowned for its high density of universities; (2) Early-year quarterly report indicates minimal revenue and a per-share loss slightly less than one dollar; (3) Share price more than doubled over a mid-2025 season, with trailing six-month total return around 80–100% by early August; (4) Develops therapeutics via a modality that directly edits DNA sequences for monogenic disorders. |
| **Testcase (Condition)** | *Headquarters in a U.S. city renowned for its high density of universities;* |
| **Testcase (Ground_truth)** | `Cambridge, Massachusetts, USA` |
| **Testcase (Condition)** | *Early-year quarterly report indicates minimal revenue and a per-share loss slightly less than one dollar* |
| **Testcase (Ground_truth)** | `Q1 2025 revenue ≈ $4.658 million; diluted EPS≈ -$0.92` |
| **Testcase (Condition)** | *Share price more than doubled over a mid-2025 season, with trailing six-month total return around 80–100% by early August)* |
| **Testcase (Ground_truth)** | `Share price rose from $1.13 (2025-04-01) to $2.51 (2025-07-31), ≈ +122%; 6-month total return as of 2025-08-01 ≈ +92%` |
| **Testcase (Condition)** | *Develops therapeutics via a modality that directly edits DNA sequences for monogenic disorders.* |
| **Testcase (Ground_truth)** | `CRISPR gene editing therapeutics (direct DNA sequence editing for monogenic diseases)` |
| **GT (Stock Symbol)** | **EDIT** |

Table 13: Example instance from the Multi-Domain.

### A.9.2    LIST OF TOOLS USED IN THE BENCHMARK

The following presents a comprehensive list of all tools used in our benchmark, categorized by their functional domain. Each tool is listed with its name and a brief description.

**Domain: `Map` (AMap Services and Google Maps Services)**

Tools for geographic information and routing services within China, powered by the AMap server.

| Tool Name | Description |
| --- | --- |
| maps_regeocode | Converts a longitude/latitude coordinate into an administrative region address. |
| maps_geo | Converts a structured address into longitude/latitude coordinates. |
| maps_ip_location | Determines the geographic location based on an IP address. |
| maps_weather | Retrieves real-time weather information for a specified city. |
| maps_bicycling_by_address | Plans a bicycle route between two locations using addresses. Unless you have a specific reason to use coordinates, it's recommended to use this tool. |
| maps_bicycling_by_coordinates | Plans a bicycle route between two coordinates. |
| maps_direction_walking_by_address | Plans a walking route between two locations using addresses. Unless you have a specific reason to use coordinates, it's recommended to use this tool. |
| maps_direction_walking_by_coordinates | Plans a walking route based on start and end longitude/latitude coordinates. |
| maps_direction_driving_by_address | Plans a driving route between two locations using addresses. Unless you have a specific reason to use coordinates, it's recommended to use this tool. |
| maps_direction_driving_by_coordinates | Plans a driving route based on start and end longitude/latitude coordinates. |
| maps_direction_transit_integrated_by_address | Plans a public transit route between two locations using addresses. Requires origin and destination city names for cross-city transit. |
| maps_direction_transit_integrated_by_coordinates | Plans a public transit route based on start and end coordinates. Requires origin and destination city names. |
| maps_distance | Measures the distance (driving, walking, or straight-line) between two coordinates. |
| maps_text_search | Searches for Points of Interest (POI) by keyword within a specified city. |
| maps_around_search | Searches for POIs near a specified coordinate and radius. |
| maps_search_detail | Retrieves detailed information for a POI by its ID. |

Tools for interacting with Google Maps for global geographic searches and directions.

| Tool Name | Description |
| --- | --- |
| googlemap_search_places | Performs a fuzzy search for places on Google Maps. |
| google_map_get_place_details | Retrieves detailed information (reviews, hours, etc.) for a specific place. |
| google_map_get_place_id | Returns the place ID(s) for places matching a search query. |
| google_map_get_map_direction | Fetches step-by-step travel directions between two locations. |

**Domain: `finance` (Financial Data Services)**

Tools for accessing stock, market, commodity, and cryptocurrency data, primarily powered by the FMP server.

| Tool Name | Description |
| --- | --- |
| get_company_notes | Gets detailed information about company-issued notes and debt instruments. |

| Tool Name | Description |
| --- | --- |
| get_income_statement | Retrieves the income statement for a company. |
| get_quote | Gets the current stock quote information. |
| get_quote_change | Gets stock price change over different time periods. |
| get_aftermarket_quote | Gets aftermarket trading quote information. |
| get_price_change | Gets price changes for a stock based on historical data. |
| search_by_symbol | Searches for stocks by ticker symbol. |
| search_by_name | Searches for stocks by company name (English only). |
| get_ratings_snapshot | Gets analyst ratings snapshot for a company. |
| get_financial_estimates | Gets analyst financial estimates for a company. |
| get_price_target_news | Gets the latest analyst price target updates. |
| get_price_target_latest_news | Gets the latest price target announcements with pagination. |
| get_company_dividends | Gets dividend history for a specific company. |
| get_dividends_calendar | Gets a calendar of upcoming dividend events for all stocks. |
| get_index_list | Gets a list of available market indices. |
| get_index_quote | Gets the current quote for a market index. |
| get_biggest_gainers | Gets a list of stocks with the biggest percentage gains. |
| get_biggest_losers | Gets a list of stocks with the biggest percentage losses. |
| get_most_active | Gets a list of most actively traded stocks by volume. |
| get_market_hours | Gets the current market hours status for a specific stock exchange. |
| get_commodities_list | Gets a list of available commodities. |
| get_commodities_prices | Gets current prices for commodities. |
| get_historical_price_eod_light | Gets historical price data for a commodity. |
| get_crypto_list | Gets a list of available cryptocurrencies. |
| get_crypto_quote | Gets current quotes for cryptocurrencies. |
| get_forex_list | Gets a list of available forex pairs. |
| get_forex_quotes | Gets the current quote for a forex pair. |
| get_ema | Gets Exponential Moving Average (EMA) values for a stock. |

**Domain: `Medical/Biology` (Biomedical Research Services)**

Tools for searching and retrieving data from biomedical databases such as PubMed, ClinicalTrials.gov, and MyVariant.info.

| Tool Name | Description |
| --- | --- |
| search | Universal search across all biomedical domains with unified query language. |
| fetch | Retrieve detailed information for any biomedical record; auto-detects domain if not provided. |
| article_searcher | Searches PubMed/PubTator3 for research articles and preprints about genes, variants, diseases, or chemicals. |
| article_getter | Fetches detailed information (abstract, full text) for a specific article by its identifier. |
| trial_searcher | Searches ClinicalTrials.gov for clinical studies based on conditions, interventions, location, etc. |
| trial_getter | Fetches comprehensive details for a specific clinical trial by its NCT ID. |
| trial_protocol_getter | Fetches core protocol information (title, summary, design, eligibility) for a clinical trial. |
| trial_references_getter | Fetches publications and references linked to a clinical trial. |

| Tool Name | Description |
|---|---|
| trial_outcomes_getter | Fetches outcome measures and results data for a clinical trial. |
| trial_locations_getter | Fetches contact and location details for sites participating in a clinical trial. |
| variant_searcher | Searches MyVariant.info for genetic variant database records (frequencies, significance, predictions). |
| variant_getter | Fetches comprehensive details for a specific genetic variant by its ID. |
| gene_getter | Get gene information from MyGene.info, including official name, aliases, genomic location and database links. |
| disease_getter | Get disease information from MyDisease.info, including definition, synonyms, ontology IDs and phenotypes. |
| drug_getter | Get drug or chemical information from MyChem.info, including structure, mechanism, indications, trade names and identifiers. |

**Domain: `web` (General Web Services)**

General-purpose tools for web searching and content parsing.

| Tool Name | Description |
|---|---|
| web_search | Performs a general web search for information. |
| web_parse | Parses a specific webpage or image URL to extract information based on a user query. |

**Domain: `Video` (Media Services)**

Tools for searching and retrieving information from YouTube.

| Tool Name | Description |
|---|---|
| google_search_images | Searches Google Images for pictures. |
| google_search_videos | Searches Google Videos for video content. |
| search_videos | Searches for YouTube videos with advanced filtering options. |
| get_video_details | Gets detailed information about a specific YouTube video. |
| get_channel_details | Gets detailed information about a specific YouTube channel. |
| get_video_comments | Retrieves comments for a specific YouTube video. |
| get_video_transcript | Retrieves the transcript for a specific YouTube video. |
| get_related_videos | Gets a list of videos related to a specific YouTube video. |
| get_trending_videos | Gets a list of trending videos on YouTube for a specific region. |
| get_video_enhanced_transcript | Advanced tool for extracting, filtering, and searching within YouTube video transcripts. |

## A.10 DISTRIBUTIONS OF QUESTION AND ANSWER LENGTH

Fig. 8 and Fig. 9 illustrate the distribution of question and ground truth answer (GT) lengths, measured in tokens, for Chinese and English questions.

Both question and answer lengths exhibit skewed distributions, with most samples concentrated in shorter ranges and a long tail extending to larger lengths. This indicates that the benchmark primarily consists of concise inputs and outputs while still including a subset of more complex, lengthier cases.

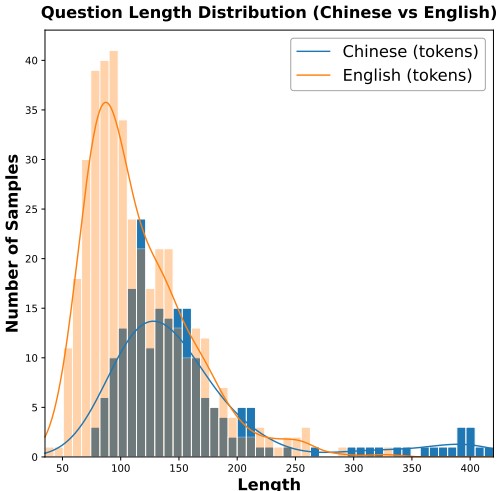

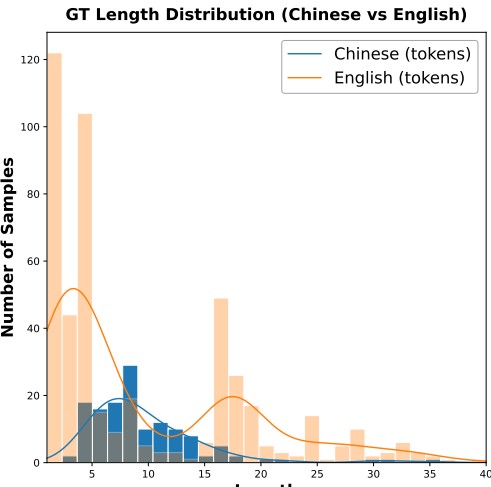

Figure 8: Question Length Distribution (Chinese vs English).

Figure 9: Ground Truth (GT) Length Distribution (Chinese vs English).

| Domain | English | Chinese |
|--------|---------|---------|
| video | 50 | 50 |
| multi-domain | 59 | 44 |
| map | 0 | 135 |
| web | 100 | 0 |
| finance | 100 | 0 |
| bio/med | 83 | 0 |
| **Total** | **392** | **229** |

Table 18: Language (English / Chinese) distribution across domains in the dataset.

## A.11 COMPREHENSIVE EVALUATION AND ANALYTICAL PROTOCOLS

### A.11.1 HUMAN STUDY

The Human Evaluation Prompt provides detailed guidelines for assessing the Factual Alignment, Coherence, and Difficult of InfoMosaic-Bench items across three evaluation dimensions.

```
What you will see per item
- Task statement (the question users must answer)
- Ground Truth
- Conditions (intermediate requirements)

Important rules
1. Use only the given materials (task/conditions/GT). You can use outside
      search or prior knowledge.
2. Judge quality of the dataset item, not the general truth of the world.

Dimension A    Factual Alignment (A n s w e r Condition  Consistency)
Question: Does the Ground Truth faithfully correspond to the stated
    conditions? Specifically, can each condition be traced to part of the
     answer, and does the answer rely only on information that is covered
     by the conditions?
How to judge:
 Check whether the conditions    Ground Truth mapping is consistent. The
      answer should:
```

```
1. Be derivable by satisfying all listed conditions.
2. Not include extra information beyond the conditions.
3. Not be solvable by ignoring some conditions.
Labels (choose one):
- A5    Fully aligned: Every condition is reflected in the final answer;
    no inconsistent conditions for Ground Truth.
- A3    Partially aligned: Answer covers the main intent, but some
    conditions are not describing the ground truth
- A1    Misaligned: Final answer does not respect conditions (
    contradicting, or irrelevant conditions).

Dimension B    Coherence (Semantic & Logical Coherence)
Question: Is the item well-formed as a dataset example? Are the task,
    conditions, and answer mutually consistent, unambiguous, and
    executable?
How to judge: Check the clarity, non-contradiction, and referential
    coherence among task    conditions    ground truth.
Labels (choose one):
- B5    Coherent: Task is clear; conditions are interpretable and non-
    conflicting; no unnatural or self-contradictory phrasing.
- B3    Minor issues: Small ambiguity, mild redundancy, or slightly
    awkward phrasing that does not block execution or interpretation.
- B1    Incoherent: Contradictory conditions, ill-formed references (e.g
    ., undefined entity), or conditions that cannot be executed with the
    given tools.
Notes:
- If a single condition directly gives away the answer, mark B1/B0
    depending on severity and explain (this will also affect Difficulty
    below).

Dimension C    Task Difficulty (Need for Multi-Tool / Multi-Step
    Reasoning)
Question: Would a competent agent need multiple tools and/or multiple
    steps to solve this task using the provided domain tools?
How to judge: Focus on necessity (not just the count of calls). If a
    single condition or a single tool suffices, the task is trivial.
Likert score (1  5 ):
- C1    Trivial: Solvable via one tool or one condition; no integration
    needed.
- C2    Easy: Mostly one source plus minimal lookup.
- C3    Moderate: Requires combining  2  pieces of evidence or
    sequential steps, but straightforward.
- C4    Hard: Clear need to aggregate multiple tools/conditions; non-
    obvious composition.
- C5    Very hard: Long-horizon integration across several tools/
    conditions; careful alignment needed.
Heuristics:
- If web-like single retrieval could answer it    C 1 C2 .
- If at least two independent conditions must be satisfied/combined
    C 3 C5  (pick based on complexity).
```

## A.12 HUMAN PERFORMANCE BASELINE STUDY

To address the reviewer request for a human-performance baseline, we conducted a controlled user study measuring human success on a random subset of INFOMOSAIC-BENCH. :contentReference[oaicite:0]index=0 All participants were instructed to solve tasks *without any AI tools* and only using standard, non-AI information sources (e.g., search engines and domain websites/apps).

### A.12.1 TASKS AND SAMPLING

We sampled 10 problems from each domain (**Map, Video, Medical/Bio, Finance, Web, Multi-domain**), resulting in 60 unique problems total. Each problem was solved by 3 different participants, yielding 180 human attempts (30 attempts per domain).

|         | Human performance | GPT-5 with web-tool | Time/min |
|---------|-------------------|---------------------|----------|
| map     | 43.33%            | 33.33%              | 8.26     |
| video   | 36.67%            | 28.33%              | 22.23    |
| bio     | 30.00%            | 48.33%              | 20.50    |
| web     | 13.30%            | 23.33%              | 25.33    |
| finance | 20.00%            | 48.33%              | 22.63    |
| multi   | 6.00%             | 45.00%              | 25.80    |
| overall | 27.89%            | 37.67%              | 0.79     |

Table 19: Human baseline performance on 60 randomly sampled INFOMOSAIC-BENCH problems (10 per domain), each attempted by 3 participants (180 attempts total) compared with performance on GPT-5 with web-tool.

### A.12.2 PARTICIPANTS

We recruited 36 participants with engineering backgrounds: 10 undergraduate students and 26 graduate students (self-reported). No domain expertise beyond general STEM training was required.

### A.12.3 PROTOCOL AND INSTRUCTIONS

Each participant completed 5 problems (total budget $\leq$ 150 minutes). Each problem had a strict time limit of **30 minutes**. Participants were required to submit the following fields per problem: **(1) Answer, (2) Time spent, (3) Tools used**, and optionally **(4) Search keywords**. If they could not solve a problem, they were instructed to report "*N/A*" ("None").

**Allowed tools.** Participants could use standard non-AI tools, including: (1) web search engines (Google/Bing/Baidu), (2) domain websites or apps (e.g., Google Maps/YouTube/Wikidata and basic databases), (3) calculators and standard browsers without AI features.

**Forbidden tools.** Participants were explicitly prohibited from using any AI assistants or AI-augmented search/browsing plugins (e.g., ChatGPT/GPT-4/Gemini/Claude/Perplexity/Copilot).

### A.12.4 SCORING

We scored human answers against the benchmark ground-truth answers. A submission was marked correct only if it matched the unique ground-truth target for the instance (e.g., exact entity/ticker/URL), consistent with the benchmark's strict end-to-end success notion.

### A.12.5 RESULTS

Table 19 reports domain-wise accuracy and average time. Overall, humans solved $50/180$ attempts correctly (**27.8%**). Map tasks were the easiest and fastest, while multi-domain and web tasks were the hardest and frequently reached the time limit.

### A.12.6 DISCUSSION

This human baseline indicates that the benchmark remains challenging even for competent STEM participants equipped with standard non-AI tools. Performance varies substantially by domain: tasks grounded in structured interfaces (e.g., maps) are comparatively more accessible, whereas open-web and multi-domain integration problems are substantially harder under a 30-minute time budget. The near-ceiling average time in Web and Multi-domain (25+ minutes) suggests that failures are often associated with long-horizon search and cross-source integration difficulty, rather than immediate lookup gaps.

### A.12.7 BENCHMARK EVALUATION PROMPT

Below are the exact evaluation prompts used to extract answers, verify correctness, and assess sub-answers for our benchmark.

**extract_answer_template**

```
You are a helpful AI assistant tasked with extracting the final answer
    from a provided solution.

**Input:**
1. A problem statement, prefixed with "===Problem: <problem>".
2. A solution to the problem, prefixed with "===Solution:".

**Problem and Solution:**
===Problem: {task}

===Solution: {operated_text}

**Instructions:**
- Carefully analyze the solution and extract the final answer in reply: "
    The answer is <answer extracted> in reply".
- If the solution does not contain a final answer (e.g., only reasoning,
    code without execution, or incomplete information), respond with: "
    The reply doesn't contain an answer."
- Ensure that the extracted answer is exactly as presented in the
    solution. Do not infer or use external knowledge. Do not execute the
    code yourself.
- Remember, Never execute the code yourself! Never doing any computation
    yourself! Just extract and output the existing answer!
```

**eval_prompt_template**

```
You are a helpful AI assistant. You will use your coding and language
    skills to verify the answer.
You are given:
  1. A problem, which is going to start like "===Problem: <problem>".
  2. A ground truth answer, which is going to start like "===Ground truth
      answer:".
  3. A reply with the answer to the problem, which are going to start
     like "===Reply:".
Please do the following:
1. Extract the answer in reply: "The answer is <answer extracted> in
    reply".
2. Check whether the answer in reply matches the ground truth answer.
    When comparison is not obvious (for example, 3*\\sqrt(6) and 7.348),
    you may compare by calculation, allowing a small margin of error.
3. After everything is done, please give each reply a comment like the
    following options:
  - "The answer is correct."
  - "The answer is approximated but should be correct. Correct Answer: <
      ground truth answer> | Answer extracted: <answer extracted>."
  - "The answer is incorrect. Correct Answer: <ground truth answer> |
      Answer extracted: <answer extracted>."
  - "The reply doesn't contain an answer."
Here are the problem, the ground truth answer and the reply:
===Problem: {task}

===Ground truth answer: {ground_truth}

===Reply: {operated_text}
```

**eval_testcase_prompt_template**

```
You are a helpful AI assistant. You will use your coding and language
    skills to verify the subanswer.
You are given:
  1. A set of subproblems and its corresponding subanswers, which are
      going to start like "===Subproblems and subanswers".
  2. A reply with the subanswer to the subproblem, which are going to
      start like "===Reply:".
Please do the following:
1. Matching the subquestions and compare the subanswer in reply with the
    ground truth subanswer.
2. Check whether the subanswer in reply matches the ground truth
    subanswer. When comparison is not obvious (for example, 3*\\sqrt(6)
    and 7.348), you may compare by calculation, allowing a small margin
    of error. Here are some principles:
- If the ground truth subanswer is a numerical value, you may compare the
     numerical value in the subanswer with the ground truth subanswer,
    allowing a small margin of error.
- If the ground truth subanswer is a string, you may compare the string
     in the subanswer with the ground truth subanswer, case-insensitive,
    and justify if it is the same as the ground truth subanswer.
3. After everything is done, please give each reply a comment like the
    following format to justify if the subanswer is correct or incorrect:
'''json
{{
"subquestion": "CORRECT|INCORRECT",
"subquestion": "CORRECT|INCORRECT"
}}
'''
Here are the problem, the ground truth answer and the reply:
===Subproblems and subanswers:
{task}

===Reply:
{subanswer}

## Output format: You must follow the following format
'''json
{{
"subquestion": "CORRECT|INCORRECT",
"subquestion": "CORRECT|INCORRECT"
}}
'''
```

