# OpenReview forum: "InfoMosaic-Bench: Evaluating Multi-Source Information Seeking in Tool-Augmented Agents"
_ICLR.cc/2026/Conference — ICLR 2026 Poster_

### Official Review · Reviewer_EoM5 · 2025-10-20

**Soundness:** 3
**Presentation:** 2
**Contribution:** 2
**Rating:** 2
**Confidence:** 4

**Summary:**

This paper introduces InfoMosaic-Bench, a new benchmark for evaluating the ability of AI agents to perform multi-source information seeking. To construct the benchmark, the paper proposes InfoMosaic-Flow, a two-stage (generation and refinement) agentic pipeline that synthesizes 621 tasks across 6 domains (bio, finance, web, maps, video, and multi-domain). The authors conducted extensive experiments on 14 models using a custom agent scaffold.

**Strengths:**

1. The paper presents an innovative attempt to generate multi-source information retrieval tasks using a two-stage agentic pipeline.

2. The work reflects a substantial effort, encompassing 621 tasks, 6 domains, 77 tools, and a comprehensive evaluation of 14 models.

3. The authors have open-sourced the code, promoting reproducibility.

**Weaknesses:**

1. The evaluation is limited to a single, custom agent scaffold. The choice of agent architecture could significantly influence performance, and this variable is not explored.

2. I have concerns about the dataset's quality. While the paper mentions quality assurance through human annotation (on a sampled subset), it lacks a human performance baseline. I am particularly skeptical about answer uniqueness and whether questions might have multiple valid answers.

3. The paper appears rushed, which affects its clarity. The writing is often difficult to follow, and the core methodology—the data synthesis pipeline—is not explained in a way that is easy to comprehend. There are also several writing issues:
	- L255: "serious" should be "series".
	- L341 (minor): "Open-sourced Model" is better described as "Open-weight Model".
	- L675: Figure 5 shows t-SNE visualization of query in different domains, but finance domain is missing.
	- L810: Table 8's example is clearly not from Bio domain as it states.
	- L829: Table 9's example is in Chinese with no English translation, while other examples are translated.

**Questions:**

1. What portion of the dataset is in Chinese? This information is not mentioned in the main paper, only alluded to in the appendix.

2. Did you conduct an ablation study where the agent has no access to any tools (including web search)? It seems some of the questions may not require any tool use to answer.

3. During the web search experiments, did you encounter any issues with rate limiting from the search engines?

4. The choice of domains seems somewhat arbitrary. What was the rationale for selecting domains as disparate as biology and finance, alongside more general ones like web and video?

5. The results for the web domain are surprising: Claude-4-Sonnet achieved only 3% accuracy, while GLM-4.5 reached 11%. Have you analyzed the reasons for this performance discrepancy?

---

> ### Author Response · Authors · 2025-11-24
> **[Part 1/2] Author Reponse**
>
> We sincerely appreciate your time and thoughtful comments. Below, we respond to each point in detail.
>
> ---
>
> **[Weakness 1]**
>
> The evaluation is limited to a single, custom agent scaffold. The choice of agent architecture could significantly influence performance, and this variable is not explored.
>
> **Answer**:
>
> 1. To examine whether our results depend on the chosen architecture, we additionally evaluated a retrieval-driven agent scaffold based on search-o1 [1] (DeepSeek-R1)
> 2. These results show the same qualitative pattern as in our main experiments:
>   1.  (i) adding domain tools yields substantial gains across all domains, and
>   2.  (ii) multi-domain tasks remain significantly more challenging than single-domain ones.
> This indicates that our benchmark conclusions are not tied to a single architectural choice.
>
> | **tool**        | **Map** | **Medical/Bio** | **Video** | **Web** | **Finance** | **Multi-domain** |
> |-----------------|---------|-----------------|-----------|---------|-------------|------------------|
> | **-**           | 0%      | 0%              | 0%        | 4%      | 19%         | 4.05%            |
> | **Domain tool** | 45.26%  | 1.56%           | 25%       | 10.5%   | 34%         | 30%              |
>
> ---
>
> **[Weakness 2-1]**
>
> Concerns about the dataset's quality. While the paper mentions quality assurance through human annotation (on a sampled subset), it lacks a human performance baseline.
>
> **Answer**:
>
> We thank the reviewer for raising this important point and clarify two aspects:
>
> 1. **Human quality control.**
>     - We conducted human quality control for all samples. As stated in L.267–270, every released instance in InfoMosaic-Bench is manually screened and refined by annotators on top of the automatic checks;
>     - The human study in the appendix is only a controlled experiment on a sampled subset to quantify the gain from this refinement, not the only place where humans are involved.
> 2. **Human performance baseline.**
>   We agree that a human performance baseline would further clarify the difficulty of the benchmark. Due to time constraints, we did not include a full human-accuracy study in the initial submission, but we are running a small-scale human evaluation on a random subset (with the same tool access as agents) and will report this human accuracy as a baseline in the camera-ready version.
>
>
> ---
>
> **[Weakness 2-2]**
>
> I am particularly skeptical about answer uniqueness and whether questions might have multiple valid answers.
>
> **Answer**:
>
> We address this using our testcase–pass-rate evaluation shown in table below. Since each query is paired with multiple hidden testcases and a GT answer, a model prediction that satisfies all constraints across testcases yields a pass rate of 1. If multiple valid answers existed, we would observe many cases with **pass = 1 but accuracy = 0**.
>
> For example in GPT-5 inference with web tool, only a few instances fall into this category : e.g., **2 over 135** in Map, **2 over 83** in Medical/Bio, 0 in all other domains. (We conduct manual inspection and confirm these non 0 items are due to minor checker tolerance issues, not genuinely different correct answers, making answer non-uniqueness is negligible in practice.)
>
> |                      | **Map** | **Medical/Bio** | **Video** | **Web** | **Finance** | **Multi-domain** |
> |----------------------|---------|-----------------|-----------|---------|-------------|------------------|
> | **GPT5**             | 2/135   | 2/83            | 0         | 0       | 0           | 0                |
> | **o3**               | 2/135   | 2/83            | 0         | 0       | 0           | 0                |
> | **gemini-2.5-flash** | 0       | 1/83            | 0         | 0       | 0           | 0                |
> | **claude-4-sonnet**  | 2/135   | 2/83            | 0         | 0       | 0           | 0                |
> | **deepseek-v3-a22b** | 1/135   | 3/83            | 0         | 0       | 0           | 0                |
> | **qwen3-235b-a22b**  | 0       | 0               | 0         | 0       | 0           | 0                |
> | **qwen3-32b**        | 0       | 1/83            | 0         | 0       | 0           | 0                |
>
>
> ---
> **[Weakness 3]**
>
> The paper appears rushed, which affects its clarity. The writing is often difficult to follow, and the core methodology—the data synthesis pipeline—is not explained in a way that is easy to comprehend.
>
> **Answer**:
>
> We sincerely thank the reviewer for their meticulous reading and constructful comments. We have carefully reviewed and corrected every instance mentioned which are highlighted in the revised version.

---

> ### Author Response · Authors · 2025-11-24
> **[Part 2/2] Author Reponse**
>
> **[Question 1]**
>
> What portion of the dataset is in Chinese? This information is not mentioned in the main paper, only alluded to in the appendix.
>
> **Answer**:
>
> We thank the reviewer for this constructive comment and address the concern in two parts:
> 1. Clarification of language composition.
>     - InfoMosaic-Bench contains 621 instances across six domains. Among these, 229 queries are written in Chinese and 392 in English, corresponding to 36.9% Chinese and 63.1% English, respectively.
>     - The evaluation pipeline (agent framework, tools, and metrics) is identical for both languages, as described in Sec. 5.1.
> 2. We have added this information in  Sec. 4 (“Dataset”), and add a domain-wise statistics in Appendix A.10.
>
>
>
>
> ---
> **[Question 2]**
>
> Did you conduct an ablation study where the agent has no access to any tools (including web search)? It seems some of the questions may not require any tool use to answer.
>
> **Answer**:
>
> We conduct experiments on agents without tools across domains on GPT-5, and find that only 6.77% samples could be solved without tools.
>
> |               | **Map** | **Video** | **financial** | **medical/bio** | **web** | **multi-domain** | **Avg** |
> |---------------|---------|-----------|---------------|-----------------|---------|------------------|---------|
> | **w.o. tool** | 2.92%   | 2.94%     | 10.59%        | 7%              | 9%      | 9.46%            | 6.77%   |
> | **w. tool**   | 32.59%  | 36.00%    | 41.00%        | 53.10%          | 29.00%  | 41.75%           | 38.18%  |
>
>
> ---
> **[Question 3]**
>
> During the web search experiments, did you encounter any issues with rate limiting from the search engines?
>
> **Answer**:
>
> No, we did not encounter rate‑limiting issues during the web‑search experiments.
>
> Our setup used SerperAPI, which provides a throughput of approximately 300 queries per second, and we restricted evaluation to a parallelism level of 50 samples. This is far below the provider’s limit, and thus we observed no throttling or request failures.
>
>
> ---
> **[Question 3]**
>
> The choice of domains seems somewhat arbitrary. What was the rationale for selecting domains as disparate as biology and finance, alongside more general ones like web and video?
>
> **Answer**:
>
> We thank the reviewer for raising this point and clarify our rationale for the chosen domains:
> 1. **We choose these domain mainly due to the tool readyness.** We intentionally selected domains where stable, publicly available MCP servers already exist , so that tasks can be grounded in real tools rather than synthetic APIs. This is also why could be seen that many other Benchmarks also tests on these domains.
> 2. **We also consider tools based on their heterogeneous information types.** Together they cover structured numeric/tabular data (finance, medical), geo-spatial information (maps), multimedia content (video), and unstructured text (web), which is essential for testing genuine multi-source integration.
>
>
>
> [1] Li, Xiaoxi, et al. "Search-o1: Agentic search-enhanced large reasoning models." arXiv preprint arXiv:2501.05366 (2025).
>
> **Finally:** We really appreciate your insightful suggestions, which have already helped us strengthen both the analysis and presentation of this work. We hope the above clarifications resolve the concerns. If the response is helpful, we would be grateful if you could consider adjusting your score accordingly. We are committed to incorporating all promised additions in the camera‑ready version to make the paper as valuable as possible to the community.

---

> > ### Comment · Reviewer_EoM5 · 2025-11-25
> >
> > Appreciate the answers to all questions. I believe some of the weakness has been addressed. I have raised the score.

---

> ### Author Response · Authors · 2025-11-25
>
> Thank you for your thoughtful review and for taking the time to consider our responses. We are highly encouraged that you raised the score and noted that "some of the weakness has been addressed."
>
> Given that the decision is still a "broad line reject", we respectfully ask if you could point us toward the most crucial, remaining flaws that must be resolved to meet the standard. Any concrete advice on the specific remaining issues would be invaluable.
>
> Thank you again for your time and insights.

---

> > ### Comment · Reviewer_EoM5 · 2025-11-25
> >
> > I am open to adjusting the score based on the outcome and analysis of the proposed experiment.
> >
> > > we are running a small-scale human evaluation on a random subset (with the same tool access as agents) and will report this human accuracy as a baseline in the camera-ready version.

---

> > > ### Author Response · Authors · 2025-12-03
> > > **Response by Authors**
> > >
> > > Thank you for the suggestion. We have now completed the proposed human-performance baseline study and added it to **Appendix A.12**.
> > >
> > > In brief, we recruited 36 engineering participants (10 undergraduate, 26 graduate), sampled 10 questions per domain (60 in total), and collected 3 independent attempts per question (180 trials in total). Each trial had a 30-minute cap, and participants were strictly prohibited from using any AI research tools, relying only on standard non-AI resources (e.g., search engines, Google Maps/YouTube/Wikidata, etc.). Overall, humans solved 50/180 trials (27.8%) correctly; human performs best in Map  (43.3%) and Video (36.67%).
> > >
> > > |  | **Human performance** | **GPT-5** | **Avg. human time (min)** |
> > > |---|---|---|---|
> > > | **map** | 43.33% | 33.33% | 8.26 |
> > > | **video** | 36.67% | 28.33% | 22.23 |
> > > | **bio/med** | 30.00% | 48.00% | 20.50 |
> > > | **web** | 13.30% | 23.33% | 25.33 |
> > > | **finance** | 20.00% | 51.66% | 22.63 |
> > > | **multi-domain** | 6.00% | 45.00% | 25.80 |
> > > | **overall** | **27.89%** | **38.27%** | **20.79** |

---

### Official Review · Reviewer_w89X · 2025-10-31

**Soundness:** 3
**Presentation:** 3
**Contribution:** 3
**Rating:** 4
**Confidence:** 3

**Summary:**

- This paper introduces InfoMosaicBench – an evaluation benchmark to assess the abilities of LLM agents to solve information-seeking tasks by gathering evidence from multiple domains/sources. The paper proposes InfoMosaic-Flow pipeline to synthetically curate their data.
- Furthermore, the paper benchmarks 14 LLMs in an agentic scaffold with only web-search and report poor task performance, highlighting the importance of domain-specific tools for multi-source information seeking queries. Even when given domain-specific tools, the performance improvements are inconsistent and achieved only for 3/6 domains.

**Strengths:**

- The analyses are very insightful and provide more nuanced insights into various failure modes of agentic systems.
- The experiments are very comprehensive and the hypotheses are thoroughly tested to empirically validate their claims.

**Weaknesses:**

- Limited real-world grounding: Are the tasks in this benchmark actually representative of real-world multi-source information-seeking tasks? The examples given in Appendix A.9.1 seem extremely complex (and somewhat obvious that they were synthetically crafted), and I am not very convinced that any human user would actually craft such complicated queries for this task.
- Understanding the choice of tools: How do authors decide which tools to include/not include for each domain? The provided list in the appendix shows extremely fine-grained tools which are only likely to be available in extremely specialized agents designed for a particular task/domain. Do these tools have any bias on the nature of tasks you synthetically curate from your InfoMosaicFlow pipeline?
- Consistency of claims: The results in Table 3 do not support the overall claim about agents with web search being incapable of solving these tasks. If that was indeed true, domain-specific tools would yield consistent improvements across all 6 domains which is not the case. Thus, is the poor task performance arising from the absence of domain-specific tools or is it just because of the overall difficulty and complexity of the task?
- Lack of insights for future work: While the paper provides an analysis of failure modes, there is limited novel insight into further agent design.

**Questions:**

Please address each of the weaknesses.

---

> ### Author Response · Authors · 2025-11-24
> **[Part 1/2] Author Response**
>
> We sincerely appreciate your time and thoughtful comments. Below, we respond to each point in detail.
>
> ---
>
> **[Weakness 1]**
>
> Limited real-world grounding: would any human user would actually craft such complicated queries for this task?
>
> **Answer**:
>
> Yes, the tasks in InfoMosaic-Bench are **derived from realistic high-stakes scenarios** that users already ask about in practice, such as searching for places under multiple constraints (e.g., time, distance, and budget), finding videos based on specific content requirements, and discovering financial companies that satisfy user-defined risk and screening constraints.
>
> At the same time, InfoMosaic-Bench is **intentionally positioned as a frontier benchmark**: as LLMs and tool ecosystems improve, users are expected to ask more complex, multi-condition questions that go beyond what current search engines and assistants handle today. The resulting query complexity is comparable to recent high-difficulty browsing/agent benchmarks (e.g., BrowseComp[1]), and is meant to reflect not only current but also emerging real-world information-seeking behaviors.
>
> ---
>
> **[Weakness 2-1]**
>
> How do authors decide which tools to include/not include for each domain
>
> **Answer**:
>
> We appreciate the reviewer’s question. Our tool selection follows three criteria:
>
> 1. **Grounded and accurate backends.**
> We only include tools that are backed by accurate and stable databases or services (e.g., production map services, biomedical registries), rather than synthetic or benchmark-specific tools. This ensures that tool outputs reflect real, verifiable facts, and every task in InfoMosaic-Bench is grounded in actual data rather than ad-hoc responses.
> 2. **Capability coverage with a minimal tool subset.**
> For each domain, we first enumerate the core capabilities we want to evaluate and then select the minimal subset of MCP tools that collectively covers these capabilities. Endpoints that are redundant with others, too narrow, or outside this capability set are not included. This keeps the tool space realistic but avoids inflating it with unnecessary or overly specialized tools.
> 3. **Operational stability and reproducibility for the community.**
> To make InfoMosaic-Bench remain runnable and reproducible over time, we intentionally choose MCP servers with stable service, reasonable rate limits, and good support for concurrent usage. For example, some prior MCP-based benchmarks[2][3] rely on Yahoo Finance MCP servers, which quickly ran into rate limits and access restrictions once widely used after release. In contrast, we use the FMP MCP server, which exposes a dedicated financial API and database and has been much more reliable under load. This makes it easier for the community to run our benchmark at scale without being blocked by backend instability.
>
> ---
>
> **[Weakness 2-2]**
>
> Do these tools have any bias on the nature of tasks you synthetically curate from your InfoMosaicFlow pipeline?
>
> **Answer**:
>
> No,the bias does not exisit on our pipeline. We will clarify our response in two parts:
>
> 1. **All tasks rely on factuality and tool-agnostic.**
>     All tasks are factual: each question is supported by verifiable outputs from the underlying services and databases. The subsequent quality-control steps (Condition Decomposing, Condition Fuzzing, and Concluding) are defined by solvability criteria—e.g., whether a single clue or a generic web lookup suffices—rather than by exploiting any particular tool’s quirks. As a result, the distribution of tasks is driven by domain semantics and multi-source requirements, not by artifacts of an individual MCP implementation.
> 2. **All MCP tools are shared with other benchmarks rather than tailored to this work.**
>     The MCP servers we use are also adopted in other MCP-centric benchmarks and tool suites. The table below shows that all of our MCP tool backends are also used in other benchmarks. This overlap indicates that our tool choices follow common practice in the MCP ecosystem, rather than a bespoke tool set constructed to favor a particular type of task.
>
> |    | **domain** | **Service provider**       | **MCP-Universe[2]** | **MCP-Bench [3]** | **MCPToolBench++[4]** | **MCP-Flow[5]** |
> |---|----|---|---|---|----|---|
> | **AMap**        | map        | Gaode Map    |   |                   | ✅                     | ✅               |
> | **Google Maps** | map        | Google Map                 | ✅                   | ✅                 | ✅                     |                 |
> | **FMP**         | financial  | FMP |                     |                   |                       | ✅               |
> | **BioMCP**      | bio/med    | PubMed/ClinicalTrials .etc |                     | ✅                 |                       |                 |
> | **YouTube**     | youtube    | Youtube |    |  |                       | ✅               |
> | **SepAPI**      | web        | Google | ✅                   |                   | ✅                     |                 |

---

> ### Author Response · Authors · 2025-11-24
> **[Part 2/2] Author Reponse**
>
> **[Weakness 3]**
>
> Consistency of claims: The results in Table 3 do not support the overall claim about agents with web search being incapable of solving these tasks. If that was indeed true, domain-specific tools would yield consistent improvements across all 6 domains which is not the case. Thus, is the poor task performance arising from the absence of domain-specific tools or is it just because of the overall difficulty and complexity of the task?
>
> **Answer**:
>
> We thank the reviewer for this careful observation. In short: the poor performance is **both** due to the absence of domain-specific tools and the overall difficulty. We will explain the inconsistent imporvements across 6 domains in the following two points:
>
> - **Tools are effective in most domains**(Table 3).
>  Table 3 shows that adding domain tools to GPT‑5 brings clear improvements over the web-only setting in multiple domains:
>   - Map: 32.59 → 40.00 (+7.41)
>   - Video: 36.00 → 46.00 (+10.00)
>   - Web: 8.00 → 11.00 (+3.00)
>
>
> - **Inconsistent gains in two domains reflect tool-exploitation bottlenecks, not tool irrelevance**(Table 3 + Fig. 4(a)).
>   Table 3 also shows accuracy drops in some domains (e.g., Medical/Bio 53.10 → 43.37, −9.73; Finance 41.00 → 30.00, −9.00). Our analysis in Fig. 4(a) illustrates these reason:
>   - Bio / Multi-domain have the **largest number of parameters** (68 / 139) and the **highest tool-usage error rates** (26.4% / 19%).
>   - Finance / Multi-domain have the **largest number of tools** (29 / 77) and the **highest tool-selection error rates** (19.1% / 11.8%).
>   This suggests that large, complex tool sets and large parameters make it difficult for current agents to choose and parameterize tools correctly, which can overshadow the potential benefits of having domain tools.
>
> ---
> **[Weakness 4]**
>
> Lack of insights for future work: While the paper provides an analysis of failure modes, there is limited novel insight into further agent design.
>
> **Answer**:
>
> We agree that making the design implications more explicit can increase the utility of the benchmark. Beyond reporting failure rates, our analyses in the main text and appendix already reveal three concrete directions for agent design. We will summarize these more clearly in a short "Implications for Agent Design" subsection.
> 1. **Tool selection and pruning are necessary when the tool space and parameter schemas are large.**
>
>     Figure 4(a) shows that in domains with many tools and complex parameter schemas (e.g., Bio, Finance, Multi-domain), the proportion of tool-usage/selection errors increases substantially. This means that simply exposing a large tool set degrades reliability. So they need to choose a small, task-specific subset of tools to improve tool calling success and avoid wasting calls on unrelated tools.
> 2. **Agents must manage their context to mitigate the impact of useless tool calls.**
>
>     Figure 3 shows that accuracy and pass rate improve as the number of tool calls increases, but only up to roughly 6–8 calls; beyond this point, performance plateaus or even decreases, while the context length keeps growing almost linearly. This indicates that indiscriminately calling more tools leads to redundant or conflicting evidence that hurts reasoning. Future agents therefore need summarize, filter, and compress intermediate tool outputs, instead of appending all raw results to the context.
> 3. **For information-seeking tasks, the main bottleneck is broad, exploratory search rather than final-step reasoning.**
>
>    Step-level analysis (Appendix A.8.1) and Figure 4(b) show that most errors occur during early, broad-search stages (e.g., retrieval miss, poor candidate selection), rather than in the final logical combination. This indicates that agents need stronger support for exploratory search and query planning over deeper reasoning based on fixed evidence.
>
> [1] Wei, Jason, et al. "Browsecomp: A simple yet challenging benchmark for browsing agents." arXiv preprint arXiv:2504.12516 (2025).
> [2] Luo, Ziyang, et al. "Mcp-universe: Benchmarking large language models with real-world model context protocol servers." arXiv preprint arXiv:2508.14704 (2025).
> [3] Wang, Zhenting, et al. "Mcp-bench: Benchmarking tool-using llm agents with complex real-world tasks via mcp servers." arXiv preprint arXiv:2508.20453 (2025).
> [4] Fan, Shiqing, et al. "Mcptoolbench++: A large scale ai agent model context protocol mcp tool use benchmark." arXiv preprint arXiv:2508.07575 (2025).
> [5] Wang, Wenhao, et al. "MCP-Flow: Facilitating LLM Agents to Master Real-World, Diverse and Scaling MCP Tools." arXiv preprint arXiv:2510.24284 (2025).
>
> ---
>
> **Finally:** We really appreciate your insightful suggestions, which have already helped us strengthen both the analysis and presentation of this work. We hope the above clarifications resolve the concerns. If the response is helpful, we would be grateful if you could consider adjusting your score accordingly.

---

### Official Review · Reviewer_hZCF · 2025-11-01

**Soundness:** 3
**Presentation:** 3
**Contribution:** 3
**Rating:** 6
**Confidence:** 3

**Summary:**

The paper presents InfoMosaic-Bench, a benchmark of 621 problems across six domains designed to test whether agents can combine general web search with domain-specific tools exposed via MCP. Tasks are synthesized with a novel pipeline called InfoMosaic-Flow, an organizer–workers pipeline that grounds conditions in verified tool outputs and iteratively prunes web-search shortcuts. Evaluating 14 LLM agents, the authors find: (i) web-only agents underperform, (ii) domain tools help selectively (not uniformly), and (iii) 22.4% of failures stem from tool selection/usage errors.

**Strengths:**

1. Originality. First benchmark that systematically evaluates multi-source information seeking with real MCP tools, bridging a gap between web-only search benchmarks and API-centric invocation tests. Organizer–workers InfoMosaic-Flow is a thoughtful synthesis design that enforces cross-source dependencies.

2. Quality. Clear two-stage generation with web-evolving verification and automated + manual quality control; ablations show verification is necessary to remove trivial cases. Well-scoped metrics (Accuracy, Pass Rate) and broad model coverage; analyses include scaling with tool calls and error taxonomies. Human study also supports data quality and reliability).

3. Significance. Results quantify a core limitation of current agents: integrating domain tools with web search; even top closed models underperform on multi-source tasks, signaling an impactful research direction.

**Weaknesses:**

1. Per-domain scale is modest. With 621 items across 6 domains (e.g., 83 medical, 100 finance/video), per-domain statistics can be noisy. Ideally should expand to a larger dev/test split per domain.

2. Judge dependence. Accuracy uses an LLM judge, which can introduce bias. Lack inter-judge agreement, including rule-based scorers where feasible (e.g., exact match for structured answers), and publish judge prompts.

3. Tool-budget sensitivity. Tool call limit fixed at K=20; conclusions may change with higher budgets or adaptive stopping. Lack sensitivity curves for K and latency/throughput trade-offs.

**Questions:**

1. How strict is the multi-source requirement post-refinement? Could you quantify the fraction of items where each condition is necessary and show a “single-source solvability” audit on a held-out judge?

2. Human study details. Could you share rating guidelines and per-domain κ? Also, does κ remain high on borderline cases that web-search could partially solve?

3. Latency & cost reporting. Any measurements of wall-clock, tokens, or tool latency per domain—and how those change under domain-tool vs. web-only settings?

---

> ### Author Response · Authors · 2025-11-24
> **[Part 1/2] Author Response**
>
> We sincerely appreciate your time and thoughtful comments. Below, we respond to each point in detail.
>
> ---
>
> **[Weakness 1]**
>
> Per-domain scale is modest. With 621 items across 6 domains (e.g., 83 medical, 100 finance/video), per-domain statistics can be noisy. Ideally, they should expand to a larger dev/test split per domain.
>
> **Answer**:
>
> We thank the reviewer for raising this concern and clarify two aspects:
> 1. **Dataset scale relative to existing MCP/tool benchmarks**.
>     Although each domain contains 83–135 items, the overall size (621 items) is already substantially larger than prior tool-agent benchmarks such as **MCP-Universe (231)** and **LiveMCPBench (95)**. Compared with these widely used evaluations, InfoMosaic-Bench offers both broader domain coverage and a significantly larger total sample count.
>
> 2. **We design the dataset to this size is due to cost constraints.**
>     Expanding per-domain size further is non-trivial because each item requires multi-step tool execution and GPT-5 verification; the table below shows the total cost in a full evaluation run, which costs $234, this cost limits the benchmark size.
>
> |                 | **Samples** | **Total Tokens** | **Tool Calls** | **Total Cost ($)** |
> |-----------------|-------------|------------------|----------------|--------------------|
> | **bio**         | 83          | 10.53M           | 1009           | 21.00              |
> | **finance**     | 100         | 21.86M           | 1477           | 44.65              |
> | **map**         | 135         | 26.24M           | 2163           | 49.42              |
> | **multidomain** | 103         | 21.17M           | 1898           | 52.65              |
> | **video**       | 100         | 15.07M           | 1843           | 31.95              |
> | **web**         | 100         | 15.39M           | 1642           | 34.67              |
> | **Total**       | 621         | 110.26M          | 10,032         | 234.34             |
>
> ---
>
> **[Weakness 2]**
>
> Accuracy uses an LLM judge, which can introduce bias. Lack inter-judge agreement, including rule-based scorers where feasible (e.g., exact match for structured answers), and publish judge prompts.
>
> **Answer**:
>
> 1. **The choice of LLM judge is due to the robustness compared to exact match. This reflects in two parts**:
>       - *Multilingual phrasing*. Exact match is brittle under multilingual phrasing, aliases, formatting variation, and numbers with units, which would incorrectly penalize many InfoMosaic answers.
>       - *Numerical and structured answers*. Our judge prompt (Appendix A.11.2) explicitly instructs the judge to perform numeric comparisons with tolerance and to normalize units/formatting, providing more reliable scoring than raw string equality.
> 2. **Rule-based scoring where feasible.**  Since our evaluation also reports PassRate (aimed to evalute parital correctness), which checks condition-by-condition correctness; this fine-grained scoring cannot be implemented reliably with exact match.
> 3. **Full transparency of judge prompts**. All judge prompts are fully published in Appendix A.11.2 and in the anonymous repository, ensuring reproducibility and allowing external verification.
>
>
>
> ---
>
> **[Weakness 3]**
>
> Tool call limit fixed at K=20; conclusions may change with higher budgets or adaptive stopping. Lack sensitivity curves for K and latency/throughput trade-offs.
>
> **Answer**:
>
> We thank the reviewer for raising this point and clarify our response in two parts:
>
> 1. **Sensitivity of accuracy to tool-call budgets.**
> Figure 3(a) in the paper reports the accuracy curve as we increase the maximum allowed tool calls. We observe that accuracy grows rapidly in the low-budget regime but **saturates after ~16 calls**. Based on this saturation behavior, we set the default budget to $K = 20$, ensuring that the agent is not artificially limited while avoiding unnecessary calls.
> 2. **Latency–accuracy trade-offs.**
> Table below (evaluation-time analysis) shows the trade-off between accuracy and execution time as the tool-call budget increases. Since improvements beyond 16 calls are minimal while latency grows substantially, $K = 20$ provides the best balance between performance and cost.
>
> |             | **Toolcalls** | **2**  | **4**  | **8**  | **16** | **32** |
> |-------------|---------------|--------|--------|--------|--------|--------|
> | **Bio/Med** | Time(s)       | 63.62  | 96.81  | 154.61 | 253.71 | 307.19 |
> |             | Acc           | 18.07% | 25.30% | 28.92% | 27.71% | 27.71% |
> | **Video**   | Time(s)       | 44.45  | 83.86  | 170.05 | 365.12 | 505.17 |
> |             | Acc           | 10.00% | 13.00% | 19.00% | 26.00% | 23.00% |
> | **Finance** | Time(s)       | 42.52  | 75.80  | 156.42 | 302.34 | 390.31 |
> |             | Acc           | 13.00% | 18.00% | 19.00% | 25.00% | 26.00% |

---

> ### Author Response · Authors · 2025-11-24
> **[Part 2/2] Author Reponse**
>
> **[Question 1]**
>
> How strict is the multi-source requirement post-refinement? Could you quantify the fraction of items where each condition is necessary and show a "single-source solvability" audit on a held-out judge?
>
> **Answer**:
>
> We thank the reviewer for this question. To directly test how necessary the annotated conditions are, we sampled 44 Map-domain instances that GPT-5 answers correctly when all conditions are present, and for each instance constructed reduced queries that keep only 1, 2, 3, or 4 conditions. Re-evaluating GPT-5 on these reduced versions, we observed that accuracy drops from 100% with all conditions to 0% for all reduced-condition variants, while the pass rate under the relaxed constraints remains relatively high (68.75%–90.91%). This indicates that, **for these audited cases, removing any subset of conditions makes the task under-specified for recovering the original gold answer**, i.e., the full set of conditions is jointly necessary in practice rather than containing redundant constraints.
>
> | **Num of Conditions per sample** | **1**  | **2**  | **3**  | **4**  | **all** |
> |----------------------------------|--------|--------|--------|--------|---------|
> | **Num samples**                  | 44     | 44     | 38     | 24     | 44      |
> | **Acc**                          | 0      | 0      | 0      | 0      | 100%    |
> | **passrate**                     | 88.64% | 90.91% | 84.21% | 68.75% | 85.95%  |
>
> ---
> **[Question 2-1]**
>
> Human study details. Could you share rating guidelines and per-domain $\kappa$?
>
> **Answer**:
>
> Rating Guidelines: We have provided detailed rating guideline in Appendix A.11.1
> Perdomain $\kappa$: The table below shows a per-domain $\kappa$. These values fall in the moderate to substantial agreement range and are consistent with typical $\kappa$ levels for complex, multi-condition reasoning tasks.
>
> |   | **Map** | **Medical/Bio** | **Video** | **Web** | **Finance** | **Multi-domain** |
> |---|---------|-----------------|-----------|---------|-------------|------------------|
> | $\kappa$ | 0.930   | 0.919           | 0.980     | 0.900   | 0.901       | 0.975            |
>
> ---
> **[Question 2-2]**
>
> Also, does $\kappa$ remain high on borderline cases that web-search could partially solve?
>
> **Answer**:
>
> In our double-annotated subset (120 instances), 45 items fall into this "partially solvable by web search (GPT-5)" category. We computed κ specifically on this subset and obtained 0.900, which is close to the overall agreement level ($\kappa=0.92$). This indicates that borderline cases do not introduce noticeably higher annotation variance.
>
>
> ---
> **[Question 3]**
>
> Latency & cost reporting. Any measurements of wall-clock, tokens, or tool latency per domain—and how those change under domain-tool vs. web-only settings?
>
> **Answer**:
>
> We conduct statistical analysis on GPT-5 evaluation. The tables blow show the Avgerate token, average tool-calls, average wall-clock per sample and average tool latency in each domain with web-only and domain-tool respectively.
>
> | _web-tool_                | **Map** | **Medical/Bio** | **Video** | **Web** | **Finance** | **Multi-domain** |
> |--------------------------|---------|-----------------|-----------|---------|-------------|------------------|
> | **Avg tokens**           | 194.4k  | 126.9k          | 150.7k    | 153.9k  | 218.6k      | 205.5k           |
> | **Avg tool-calls**       | 16.02   | 12.15           | 18.43     | 16.42   | 14.77       | 18.42            |
> | **Avg wall-clock (s)**   | 411.6   | 306.6           | 504.6     | 658.8   | 390.0       | 920.4            |
> | **Avg tool latency (s)** | 1.53    | 1.68            | 1.48      | 2.03    | 2.00        | 2.33             |
>
>
> | _domain-tool_            | **Map** | **Medical/Bio** | **Video** | **Web** | **Finance** | **Multi-domain** |
> |--------------------------|---------|-----------------|-----------|---------|-------------|------------------|
> | **Avg tokens**               | 124.9k  | 130.3k          | 112.8k    |  208.3k | 177.3k      | 214.6k           |
> | **Avg tool-calls**           | 15.77   | 10.03           | 14.28     | 18.99   | 16.60       | 19.39            |
> | **Avg wall-clock (s)**   | 405.6   | 253.1           | 391.0     | 637.87  | 438.3       | 986.8            |
> | **Avg tool latency (s)** | 1.01    | 3.19            | 2.11      | 1.62    | 1.31        | 2.36             |
>
>
> ---
>
> **Finally:** We really appreciate your insightful suggestions, which have already helped us strengthen both the analysis and presentation of this work. We hope the above clarifications resolve the concerns. If the response is helpful, we would be grateful if you could consider adjusting your score accordingly. We are committed to incorporating all promised additions in the camera‑ready version to make the paper as valuable as possible to the community.

---

### Official Review · Reviewer_S6ix · 2025-11-01

**Soundness:** 3
**Presentation:** 3
**Contribution:** 3
**Rating:** 6
**Confidence:** 4

**Summary:**

The authors introduce a benchmark to test multi-source information seeking by tool-augmented LLM agents. It comprises 621 query tasks spanning six domains and provides 77 specialized MCP tools for these domains. Each task requires combining general web search with one or more domain-specific tool calls. Tasks are automatically synthesized via a pipeline, which uses an “organizer-workers” LLM architecture to propose coherent scenarios grounded in verified tool outputs. Evaluation results on 14 LLMs  results show that web search alone yields very low accuracy. Adding domain tools only yields selective gains (improving results in map/video but degrading others) and about 22.4% of failures were traced to incorrect tool selection or usage.

**Strengths:**

- InfoMosaic-Bench explicitly requires agents to integrate evidence from both web search and diverse tools. No existing benchmark systematically tests this combination.
- Evaluations cover 77 MCP tools across 6 different domains.
- Tasks are explicitly designed so that no single tool or search can solve them in one step.

**Weaknesses:**

- Because all tasks are synthetically generated, the dataset may not accurately reflect the real distribution of human information-seeking queries.
- Some conclusions may over-generalize. The reported improvement from tool use in map and video tasks, and degradation in medical and financial ones, likely reflects differences in the specific tools provided and the complexity of their parameterization rather than inherent domain effects.
- The dataset has uneven domain sizes (e.g., 135 map vs. 90 biomedical tasks) and mixes Chinese and English, with varied text lengths and complexity. While this adds realism, it complicates cross-domain comparisons and may introduce language-related bias.

**Questions:**

- The current evaluation is binary at the task level. Would it be feasible to design a metric that captures partial correctness?

---

> ### Author Response · Authors · 2025-11-24
> **[Part 1/2] Author Response**
>
> We sincerely appreciate your time and thoughtful comments. Below, we respond to each point in detail.
>
> ---
>
> **[Weakness 1]**
>
> Because all tasks are synthetically generated, the dataset may not accurately reflect the real distribution of human information-seeking queries.
>
> **Answer**:
>
> We thank the reviewer for raising this concern and clarify two aspects:
> 1. **The tasks in InfoMosaic-Bench are derived from realistic high-stakes scenarios that users already ask about in practice**, such as searching for places under multiple constraints (e.g., time, distance, and budget), finding videos based on specific content requirements, and discovering financial companies that satisfy user-defined risk and screening constraints.
> 2. **We have manually validated all items**. Every synthesized example is manually reviewed to ensure that (i) the query is linguistically natural, (ii) the information need is plausible for a real user of the corresponding tools, and (iii) the answer is indeed obtainable from the available tool ecosystem. Items that look contrived or unrealistic are edited or discarded.
>
> ---
> **[Weakness 2]**
>
> Some conclusions may over-generalize. The reported improvement from tool use in map and video tasks, and degradation in medical and financial ones, likely reflects differences in the specific tools provided and the complexity of their parameterization rather than inherent domain effects.
>
> **Answer**:
>
> We thank the reviewer for this insightful comment. We do not attribute the observed gains/degradations solely to inherent domain effects. Instead, we explicitly analyze the influence of **the MCP tool ecosystem available in that domain**.
>
> Our error analysis in Fig. 4(a) explicitly links performance variance to the scale and complexity of the toolsets, thus confirming the reviewer's point that tool parameterization is a key factor:
>
> - **High Complexity** $\rightarrow$ **Performance Degradation**: The domains showing the sharpest performance degradation (e.g., Medical/Bio: −9.73 and Finance: −9.00) correspond to the most complex tool ecosystems:
>     - Bio / Multi-domain tools have the **largest number of parameters** (68 / 139) and the highest proportion of usage errors (26.4% / 19%). These high error rates directly result in the observed accuracy drops.
>     - Finance / Multi-domain tools have the **largest number of tools** (29 / 77) and the highest proportion of selection errors (19.1% / 11.8%).
>
> - **Lower Complexity** $\rightarrow$ **Performance Gains**: Conversely, the domains showing the clearest performance improvements (e.g., Map: +7.41 and Video: +10.00) correspond to toolsets with moderate complexity. For instance, Map tools achieve the highest proportion of valid results (26.1%)—this low error rate is why the tool's benefit can successfully translate into a stable performance gain.
>
>
> ---
>
> **[Weakness 3]**
>
> The dataset has uneven domain sizes and mixes Chinese and English, with varied text lengths and complexity. While this adds realism, it complicates cross-domain comparisons and may introduce language-related bias.
>
> **Answer**:
>
> We thank the reviewer for raising this concern and clarify three aspects:
>
> 1. **Per-domain reporting and sample weighting.**
>     We already report the number of samples per domain in Table 1, and all aggregate metrics are computed at the per-sample level; most of our analyses (e.g., Tables 2–3 and the radar plots) are presented per-domain, so our main conclusions do not rely on directly comparing raw accuracies across domains with very different sizes.
>
> 2. **Language distribution and text complexity.**
>     The benchmark contains both Chinese and English queries (229 vs. 392 instances), and Appendix A.10 reports length distributions of questions and answers for both languages, showing that their text complexity is broadly comparable.
>
> 3. **English-only subset analysis (to control for language effects).**
>     The table below shows that experiments on an English-only subset (at least for GPT-5 and GLM-4.5), and we have observed that the key trends—such as the insufficiency of web-only tools and the selective gains from domain-specific tools—remain qualitatively unchanged when restricting evaluation to English data.
>
> |         | **o3** | **claude-4-sonnet** | **Gemini** | **qwen3-32b** | **deepseek-v3-0324** |
> |---------|--------|---------------------|------------|---------------|----------------------|
> | **en**  | 36.44% | 14.18%              | 7.40%      | 8.71%         | 5.97%                |
> | **cn**  | 36.34% | 18.72%              | 7.11%      | 5.94%         | 6.85%                |
> | **all** | 36.35%  | 15.94%               | 7.25%       | 7.73%          | 6.28%                 |

---

> ### Author Response · Authors · 2025-11-24
> **[Part 2/2] Author Reponse**
>
> **[Question]**
>
> The current evaluation is binary at the task level. Would it be feasible to design a metric that captures partial correctness?
>
> **Answer**:
>
> We have already designed a more fine-grained evaluation metric based on test cases, namely the **pass rate**(*Line 285-288*). This metric records whether each individual condition in a task reply is satisfied, and the results are reported in the last column of Table 2.
>
>
> ---
>
> **Finally:** We really appreciate your insightful suggestions, which have already helped us strengthen both the analysis and presentation of this work. We hope the above clarifications resolve the concerns. If the response is helpful, we would be grateful if you could consider adjusting your score accordingly. We are committed to incorporating all promised additions in the camera‑ready version to make the paper as valuable as possible to the community.

---

### Author Response · Authors · 2025-12-03
**Summary of Rebuttal**

We sincerely thank all reviewers and the committee for their time. Given the recent policy change, we summarize here the core contributions and how our responses and additional experiments address the main concerns.

**Content Summarization**

The paper introduces InfoMosaic-Bench, a multi-source, tool-grounded information-seeking benchmark that fills a gap in current evaluations, which largely focus on websearch-only QA. It constructs tasks around real MCP tools across six domains (map, video, medical/bio, finance, web, multi-domain) that require **multi-step, cross-source integration** over **reliable domain backends**, rather than relying solely on general web search.

Across a comprehensive experiments and analysis, we show that:
- web search alone is often insufficient for precise reasoning on these tasks;
- adding domain-specific tools unlocks substantial headroom in several domains;
- however, current agents struggle with tool selection/parameterization in large, complex tool spaces, leading to selective and sometimes inconsistent gains—especially on multi-domain tasks.

**Addressing concerns: data quality**

Through the additional analyses and experiments in this rebuttal, we believe we have substantively addressed data-quality-related concerns:

1. **End-to-end human quality control.**
  Every released instance passes both automatic checks and subsequent manual screening/refinement by annotators. The small human rating study in the Fig. 4(c) is an extra controlled experiment on top of this pipeline, not the only human involvement.
2. **Answer uniqueness.** Using the testcase–pass-rate evaluation, we count instances where a prediction satisfies all hidden testcases (pass rate = 1) but fails accuracy. Across models, such cases are almost not exists, indicating that answer non-uniqueness is negligible in practice.
3. **Human performance** Reviewer EoM5 requested a human-performance baseline in addition to our existing analyses. After our initial response outlining the planned study, the reviewer noted that several weaknesses had been addressed and raised the score. We have now completed the requested no-AI human baseline and report the full protocol and results in Appendix A.12.

**Addressing concerns: experimental robustness**

We also added several analyses to strengthen experimental robustness and clarify the scope of our conclusions:

1. **Architecture sensitivity** (EoM5 W1). We evaluated a retrieval-driven agent scaffold based on search-o1 (DeepSeek-R1). The qualitative pattern is consistent with our main experiments: adding domain tools improves performance across domains, and multi-domain tasks remain substantially harder, indicating that our conclusions are not tied to a single agent architecture.
2. **Tool-free ablation** (EoM5 Q2). We ran GPT-5 without any tools (no web search, no MCP tools). Only 6.77% of instances are solvable in this setting, versus 38.18% with tools, confirming that InfoMosaic-Bench is meaningfully tool-dependent rather than solvable from prior knowledge alone.
3. **Tool budget and latency/cost** (hZcF W3, Q3). We report accuracy as a function of the maximum tool-call budget and observe saturation around 16 calls. We also provide per-domain statistics of average tokens, number of tool calls, wall-clock time, and tool latency for web-only and domain-tool settings. These results characterize the latency–accuracy trade-off and motivate our default choice K=20.
4. **Language distribution and bias** (EoM5 Q1, S6ix W3). We clarify the language composition (36.9% Chinese, 63.1% English) in Table 18 and show that the evaluation pipeline is identical for both languages. We further provide English-only results (on GPT-5 and GLM-4.5) and find that the main qualitative trends remain unchanged, suggesting that language mixture is not driving the conclusions.
5. **Tool and domain selection rationale** (w89X W2, S6ix W1–W2). We detail our tool and domain choices along three axes:
    - using **stable, accurate backends** (e.g., production map services, biomedical and finantial  databases) to ensure factual grounding;
    - selecting a **minimal capability-complete subset** of MCP tools per domain to cover essential operations without inflating the tool space arbitrarily;
    - preferring MCP servers that are **operationally stable and reproducible** for the community (reasonable rate limits, concurrent usage). We also show that all our MCP backends are used in other MCP-based benchmarks (MCP-Universe, MCP-Bench, MCPToolBench++, MCP-Flow), indicating alignment with existing practice rather than a bespoke tool set.

We hope this concise summary helps the area chair assess both the core contributions of InfoMosaic-Bench and the concerns raised in the reviews have been addressed in the revised version and appendix.

---

### Meta-Review · Area_Chair_MS4s · 2026-01-02

**Summary:**

The reviewers were generally concerned about the benchmark quality, the limited evaluation with and without tools, the lack of comparison across different agent scaffolds, and the impact of the Chinese language on the benchmark. Most of these concerns were addressed in the rebuttal. However, some issues remain, including the limited diversity of agent scaffolds evaluated, the unclear significance of web search versus MCP tools, and the inclusion of Chinese-language data, which introduces additional variability into a tool-use benchmark.
I recommend acceptance because the work is timely and the community needs a strong benchmark for evaluating LLM tool use in realistic environments. I strongly encourage the authors to include a more comprehensive evaluation across different agent scaffolds and to provide a clearer separation and analysis of the Chinese portion of the benchmark.

**Reviewer Concerns:**

- Benchmark quality: Partially resolved by the authors, including clarifications on human quality control and the human performance baseline.
- Tool ablation: Partially resolved with a no-tool GPT-5 baseline.
- Choice of agent scaffold: Partially resolved with an additional scaffold based on search-o1.
- Chinese language: Partially resolved, with trends remaining similar for a subset of models.

**Reviewer Scores:**

- S6ix: Maintained the score since the concerns were partially addressed
- hZCF: Improved the score since the concerns were mostly addressed
- w89X: Maintained the score since the concerns were partially addressed
- EoM5: Improved the score since the concerns were mostly addressed

---

### Decision · Program_Chairs · 2026-01-26

Accept (Poster)